



# North Atlantic Oscillation response in GeoMIP experiments G6solar and G6sulfur: why detailed modelling is needed for understanding regional implications of solar radiation management

Andy Jones[1], Jim M. Haywood[1,2], Anthony C. Jones[3], Simone Tilmes[4], Ben Kravitz[5,6], and Alan Robock[7]

[1]Met Office Hadley Centre, Exeter, EX1 3PB, UK
[2]Global Systems Institute, College of Engineering, Mathematics and Physical Sciences, University of Exeter, Exeter, EX4 4QE, UK
[3]Met Office, Exeter, EX1 3PB, UK
[4]Atmospheric Chemistry, Observations and Modeling Laboratory, National Center for Atmospheric Research, Boulder, CO 80307, USA
[5]Department of Earth and Atmospheric Sciences, Indiana University, Bloomington, IN 47405-1405, USA
[6]Atmospheric Sciences and Global Change Division, Pacific Northwest National Laboratory, Richland, WA 99352, USA
[7]Department of Environmental Sciences, Rutgers University, New Brunswick, NJ 08901-8551, USA

*Correspondence to*: Andy Jones (andy.jones@metoffice.gov.uk)

**Abstract.** The realisation of the difficulty of limiting global mean temperatures to within 1.5 °C or 2.0 °C above pre-industrial levels stipulated by the 21[st] Conference of Parties in Paris has led to increased interest in solar radiation management (SRM) techniques. Proposed SRM schemes aim to increase planetary albedo to reflect more sunlight back to space and induce a cooling that acts to partially offset global warming. Under the auspices of the Geoengineering Model Intercomparion Project, we have performed model experiments whereby global temperature under the high forcing SSP5-8.5 scenario is reduced to follow that of the medium forcing SSP2-4.5 scenario. Two different mechanisms to achieve this are employed, the first via a reduction in the solar constant (experiment G6solar) and the second via modelling injections of sulfur dioxide (experiment G6sulfur) which forms sulfate aerosol in the stratosphere. Results from two state-of-the-art coupled Earth system models both show an impact on the North Atlantic Oscillation (NAO) in G6sulfur but not in G6solar. Both models show a persistent positive anomaly in the NAO during the Northern Hemisphere winter season in G6sulfur, suggesting an increase in zonal flow and an increase in North Atlantic storm track activity impacting the Eurasian continent leading to regional warming. These findings are broadly consistent with previous findings on the impact of stratospheric volcanic aerosol on the NAO and emphasise that detailed modelling of geoengineering processes is required if accurate impacts of SRM impacts are to be simulated. Differences remain between the two models in predicting regional changes over the continental USA and Africa, suggesting that more models need to perform such simulations before attempting to draw any conclusions regarding potential continental-scale climate change under SRM.





## 1 Introduction

Successive Intergovernmental Panel on Climate Change (IPCC) reports (*e.g.* Forster *et al*., 2007; Myhre *et al*., 2013) have highlighted that anthropogenic greenhouse gas emissions exert a strong positive radiative forcing leading to a warming of Earth's climate. However, the same IPCC reports also suggest that aerosols of anthropogenic origin exert a significant, but

poorly quantified, negative radiative forcing leading to a cooling effect on the Earth's climate through aerosol-radiation and aerosol-cloud interactions. Aerosols have therefore been at the forefront of discussions about increasing planetary albedo by deliberate injection either into the stratosphere (stratospheric aerosol interventions, SAI; Dickinson, 1996) or into marine boundary layer clouds (marine cloud brightening, MCB; *e.g.* Latham, 1990). Such putative albedo-increasing interventions are referred to as solar radiation management (SRM) geoengineering.


Initial simulations of the impacts of SAI and MCB were carried out by individual groups using models of varying complexity for a range of different scenarios, but the range of different scenarios applied to the models meant that definitive reasons for differences in model responses were difficult to establish (*e.g.* Rasch *et al*., 2008; Jones *et al*., 2010). The Geoengineering Model Intercomparison Project (GeoMIP) framework was therefore established with specific protocols for performing model

simulations under a range of defined scenarios (Kravitz *et al*., 2011). The scenarios considered by GeoMIP have themselves evolved with the earliest idealised simulations being supplemented by progressively more complex scenarios aiming to address more specific policy-relevant questions. The earliest simulations involved balancing an abrupt quadrupling of atmospheric carbon dioxide concentrations by simply reducing the solar constant (GeoMIP experiment G1; Kravitz *et al*., 2011). While such simulations are highly idealised, the simplicity of the scenario means that many climate models could perform the

simulations providing a robust multi-model assessment (Kravitz *et al*., 2013, 2020).

Policy-relevant questions regarding SRM can only be addressed by climate model simulations that represent deployment strategies which use technologies that are considered safe, cost-effective and have a reasonably short development time (Royal Society, 2009). SAI has been suggested as one such potentially plausible mechanism, its plausibility enhanced by observations

of explosive or effusive volcanic eruptions which cause a periodic negative radiative forcing and a cooling of the Earth's climate (*e.g.* Robock, 2010; Haywood *et al*., 2013; Santer *et al*., 2014; Malavelle *et al*., 2017). Observations of such natural analogues provide powerful constraints on the ability of global climate models to represent complex aerosol-radiation and aerosol-cloud processes, although the pulse-like nature of the emissions from volcanic eruptions means that they are not perfect analogues for SRM (Robock *et al*., 2013). Single model simulations which include treatments of aerosol processes associated

with SAI (*e.g.* Jones *et al*., 2017, 2018; Irvine *et al*., 2019) have shown that policy-relevant climate metrics at global, continental and regional scales such as sea-level rise, sea-ice extent, European heat waves, Atlantic hurricane frequency and intensity, and North Atlantic storm track displacement can be significantly ameliorated under SAI geoengineering compared with baseline (non-geoengineered) scenarios. Additionally, SAI strategies could potentially be tailored to provide spatial





distributions of stratospheric aerosol that mitigate some of the residual impacts of SAI such as the overcooling of the tropics
and undercooling of polar latitudes that are evident under more generic SAI strategies (*e.g.* MacMartin *et al.*, 2013; Tilmes *et al.*, 2018). However, studies suggest that SAI would by no means ameliorate all effects of climate change (*e.g.* Simpson *et al.*, 2019; Da-Allada *et al.*, 2020; Robock, 2020).

   The North Atlantic Oscillation (NAO) can be defined as a change in the pressure difference between the Icelandic low and
the Azores high pressure regions (*e.g.* Hurrell, 1995) and, by convention, a positive NAO anomaly is associated with an increase in the surface pressure gradient between these regions. Both model simulations (*e.g.* Stenchikov *et al.*, 2002) and observations (*e.g.* Lorenz and Hartmann, 2003) have shown that one of the most significant atmospheric responses following explosive volcanic eruptions is the impact on the Northern Hemisphere wintertime NAO, although the magnitude of the signal relative to natural variability has been challenged (Polvani *et al.*, 2019). Shindell *et al.* (2004) provide a concise summary of
the mechanism by which volcanic stratospheric aerosols are thought to influence the dynamical response of the NAO leading to wintertime warming over Eurasia and North America (Robock and Mao, 1992). Essentially, (1) sunlight absorbed by aerosols leads to heating of the lower stratosphere which enhances the meridional temperature gradient, (2) strengthening the westerly zonal winds near the tropopause; (3) planetary waves propagating upwards in the troposphere are refracted away from the pole due to the change in wind shear, further strengthening the westerlies; (4) the enhanced westerlies propagate down to
the surface via a positive feedback between the zonal wind anomalies and tropospheric eddies; and (5) strengthened westerly flow near the ground creates the surface pressure and temperature response patterns. As SAI geoengineering could be considered equivalent to a continuous volcanic eruption it seems plausible that it too could generate similar anomalies in the NAO and so surface temperature.

The most recent GeoMIP Phase 6 scenarios (GeoMIP6; Kravitz *et al.*, 2015) attempt to provide more policy-relevant information on SRM geoengineering by aligning with the Coupled Model Intercomparison Project Phase 6 (CMIP6; Eyring *et al.*, 2016). Two GeoMIP6 experiments will be considered here: G6solar and G6sulfur. In both experiments the modelled global-mean temperature under a high-forcing scenario is reduced to that in a medium-forcing scenario. The mechanism for performing the temperature reduction is either an idealised reduction of the solar constant (experiment G6solar) or a more
realistic injection of sulfur dioxide into the stratosphere (experiment G6sulfur) where it forms sulfate aerosol that reflects sunlight back to space. We examine results from two Earth system models which have performed both experiments, UKESM1 and CESM2-WACCM6.

   Section 2 provides a brief description of the UKESM1 and CESM2-WACCM6 models. Section 3 provides a description of
the experimental design of the G6solar and G6sulfur experiments. Results are presented in Section 4, before discussions and conclusions are presented in section 5.



## 2 Model Description

Both UKESM1 and CESM2-WACCM6 are fully coupled Earth system models which have contributed to CMIP6 and GeoMIP6. Both models (or their immediate forebears) have undergone various degrees of validation relevant to SAI using observations from explosive volcanic eruptions (*e.g.* Haywood *et al.*, 2011; Dhomse *et al.*, 2014; Mills *et al.*, 2016).

UKESM1 is described by Sellar *et al.* (2019). It comprises an atmosphere model based on the Met Office Unified Model (UM; Walters *et al.*, 2019; Mulcahy *et al.*, 2018) with a resolution of 1.25° latitude by 1.875° longitude with 85 levels up to approximately 85 km, coupled to a 1° resolution ocean model with 75 levels (Storkey *et al.*, 2018). It includes components to model tropospheric and stratospheric chemistry (Archibald *et al.*, 2020) and aerosols (Mann *et al.*, 2010), sea-ice (Ridley *et al.*, 2018), the land surface and vegetation (Best *et al.*, 2011) and ocean biogeochemistry (Yool *et al.*, 2013).

CESM2-WACCM6 is described by Danabasoglu *et al.* (2020) and Gettelman *et al.* (2019a). The atmosphere model has a resolution of 0.95° in latitude by 1.25° in longitude with 70 levels from the surface to about 140 km. This is coupled to an ocean model component with a nominal 1° resolution and 60 vertical levels (Danabasoglu *et al.*, 2012) and a sea-ice model (Hunke *et al.*, 2015). It includes a full stratospheric chemistry scheme that is coupled to the atmospheric dynamics, aerosol and radiation schemes (Mills *et al.*, 2017) and a land model with interactive carbon and nitrogen cycles (Danabasoglu *et al.*, 2020).

## 3 G6solar and G6sulfur Experimental Design

As described in Kravitz *et al.* (2015), the goal of GeoMIP experiments G6solar and G6sulfur is to modify simulations based on ScenarioMIP high forcing scenario SSP5-8.5 (O'Neill *et al.*, 2016; experiment ssp585) so as to follow the evolution of the medium forcing scenario SSP2-4.5 (experiment ssp245). Kravitz *et al.* (2015) define the criterion for comparing the modified simulations with their ssp245 target in terms of radiative forcing. This was subsequently found to be impractical for some models and so for GeoMIP6 the criterion applied was that for each decade from 2021 to 2100 the global, decadal-mean near-surface air temperature of G6solar or G6sulfur should be within 0.2 K of that of the corresponding decade of each model's ssp245 simulation. Experiment G6solar performs the required modification in an idealised manner by gradually reducing the solar constant over the 21st century, whereas G6sulfur achieves it by the arguably more technologically feasible method of injecting gradually increasing amounts of $SO_2$ into the lower stratosphere. $SO_2$ was injected continuously between 10° N - 10° S along the Greenwich meridian at 18-20 km altitude in UKESM1 and on the equator at the dateline at ~25 km altitude in CESM2-WACCM6.

The results presented are ensemble means of three (UKESM1) or two (CESM2-WACCM6) members. These are ultimately initial condition ensembles: the G6solar and G6sulfur ensemble members are based on ensemble members of each model's





ssp585 experiment, which are themselves continuations of corresponding CMIP6 historical simulations, which in turn are
initialised from different points in each model's pre-industrial control simulation.

We investigate the impact of SAI by examining differences between G6sulfur and G6solar, generally over the final 20 years of the 21st century. We are thereby comparing two experiments in which the temperature evolution is nominally the same but which achieve this by different methods. This should highlight any impacts which are captured by a more detailed treatment
of modelling SAI geoengineering (G6sulfur) which are not seen when geoengineering is treated in a more idealised fashion (G6solar).

## 4 Results

We first provide a brief analysis of the levels of success that G6sulfur and G6solar have in reducing the temperature change to that of ssp245. As the experimental design assures that the decadal mean temperature in G6sulfur and G6solar are within 0.2
K of the values for ssp245, we do not show the temporal evolution of temperature, but there is some merit in examining the inter-model and inter-forcing differences of the resulting spatial patterns of temperature change to give context to the results that follow. When analysing the results from the simulations, we generally focus on the difference 'G6sulfur minus G6solar' for several key variables that are associated with our understanding of the influence of stratospheric aerosol on the development of NAO anomalies.

### 4.1 Spatial Distribution of 21st Century Temperature Change

The spatial pattern of the global mean temperature change is calculated as the change from present day (PD; mean of 2011-2030) compared with the period 2081-2100 and is shown for experiments ssp245, G6solar and G6sulfur for UKESM1 and CESM2-WACCM6 in Fig. 1.

***Figure 1***

It is obvious from Fig. 1 that the inter-model differences in temperature response (*i.e.* the differences between the top and bottom rows) are much greater than the inter-forcing differences in temperature response (*i.e.* the differences between the columns in any one row). In UKESM1 the warming is around 2.6 K compared with present-day, while for CESM2-WACCM6
the warming is more moderate at around 1.9 K. This result is interesting in itself because the base models that are used in these simulations have been diagnosed as having equilibrium climate sensitivities (*i.e.* for a doubling of $CO_2$) of 5.4 K (UKESM1; Andrews *et al.*, 2019) and 5.3 K (CESM2; Gettelman *et al.*, 2019b); one might thus expect a similar transient climate response under the SSP2-4.5 scenario.





Both models warm over land regions more than over ocean regions as documented in successive IPCC reports (*e.g.* Forster *et al.*, 2007; Myhre *et al.*, 2013). UKESM1 shows a strong polar amplification, particularly in the Northern Hemisphere, while polar amplification is more muted in CESM2-WACCM6. This is likely linked to differences in poleward atmospheric and oceanic heat transport. Indeed, CESM2-WACCM6 suggests that areas of the North Atlantic are subject to a cooling as the mean climate warms. This is presumably as a result of a strong reduction of the Atlantic Meridional Overturning Circulation

which has been documented to collapse in CESM2 from a present-day level of ~23 Sv to ~8 Sv by 2100 under the SSP5-8.5 scenario (Muntjewerf *et al.*, 2020; Tilmes *et al.*, 2020). UKESM1 shows no such behaviour.

The similarity between the inter-forcing patterns of temperature responses in ssp245, G6solar and G6sulfur for each model is quite striking. On the basis of such an analysis it would be tempting to conclude that G6solar, which has the benefits of

being relatively simple to implement in a great number of climate models (*e.g.* Kravitz *et al.*, 2013, 2020), might be a reasonable analogue for the far more complex G6sulfur simulations. This conclusion will be examined in the following sections.

### 4.2 Stratospheric Aerosol Optical Depth

In G6sulfur the mean $SO_2$ injection rate during the final two decades (2081-2100) is 19.0 Tg yr$^{-1}$ for UKESM1 and 20.6 Tg

yr$^{-1}$ for CESM2-WACCM6. The resulting anomalies in annual mean aerosol optical depth (AOD, determined at 550 nm) for the final 20 years are 0.33 for UKESM1 and 0.28 for CESM2-WACCM6; their geographic distributions are shown in Fig. 2.

***Figure 2***

By 2081-2100 the AOD needed to reduce the SSP5-8.5 temperature levels to those of SSP2-4.5 is some 18% greater for UKESM1 than for CESM2-WACCM6, although the amount of cooling produced in the two models is very similar (-2.47 K for UKESM1 and -2.33 K for CESM2-WACCM6). This can be attributed to the different $SO_2$ injection strategies and to different transport strengths from the tropics to the poles in the Brewer-Dobson circulation of the stratosphere. In UKESM1 there is considerably more geoengineered AOD in the tropical reservoir (*e.g.* Grant *et al.*, 1996) than in CESM2-WACCM6

where the transport to higher latitudes is more efficient.

### 4.3 Stratospheric Ozone

Stratospheric aerosol is widely acknowledged to reduce stratospheric ozone through heterogeneous chemistry processes, particularly in polar regions (*e.g.* Solomon, 1999; Tilmes *et al.* 2009) and has been studied in earlier GeoMIP activities (*e.g.* Pitari *et al.*, 2014). Both UKESM1 and CESM2-WACCM6 include detailed stratospheric chemistry and are capable of

modelling the impact of stratospheric aerosol on stratospheric ozone (Morgenstern *et al.*, 2009; Mills *et al.*, 2017). The impact of SAI on stratospheric ozone concentrations is shown in Fig. 3.



***Figure 3***

The SAI-induced changes in ozone concentration between G6solar and G6sulfur are consistent with the distributions of aerosol in the two models. UKESM1, with its higher concentration of aerosol in the tropical reservoir, shows a greater tropical ozone change, with the maximum reduction centred around 20-30 hPa (~24-27 km) for both models. These changes are consistent with the findings of Tilmes *et al.* (2018) and are a combination of chemical and transport changes. The reduction in ozone concentrations in the tropics around 20-30 hPa is the result of an increase in vertical advection, while the increase in

ozone above this is a result of a decreased rate of catalytic $NO_x$ ozone loss cycle (see Tilmes *et al.*, 2018 for more details).

**4.4 Stratospheric Temperature**

Perturbations to stratospheric temperatures are a key mechanism implicated in observed and modelled changes in the northern hemispheric wintertime NAO subsequent to stratospheric aerosol injection from volcanoes (*e.g.* Stenchikov *et al.*, 2002; Lorenz and Hartmann, 2003; Shindell *et al.*, 2004). The annual-mean and the Northern Hemisphere wintertime (December-

February) stratospheric temperature perturbations are shown in Fig. 4.

***Figure 4***

For both models, the peak in the annual mean temperature perturbation is in the tropics which is where the $SO_2$ is injected

and the resulting stratospheric AOD is greatest (Fig. 2). Differences between the models' aerosol and radiation schemes means that CESM2-WACCM6 has slightly more warming in the tropical stratosphere despite having somewhat lower AOD compared with UKESM1. Although stratospheric sulfate is primarily a scattering aerosol in the solar part of the spectrum, the small degree of absorption of solar radiation by the stratospheric aerosols in the near infra-red is the primary cause of stratospheric heating (*e.g.* Stenchikov *et al.*, 1998; Jones *et al.*, 2016). Perturbations to stratospheric temperatures in the tropics due to less

ultra-violet absorption from the reduction of stratospheric ozone (Fig. 3) plays a more minor role. The right-hand panels of Fig. 4 show that the impact of solar absorption in the stratosphere cannot be effective during the polar night, thus stratospheric heating from the aerosol is only present at latitudes south of the Arctic Circle (Shindell *et al.*, 2004). The cooling at high latitudes during Northern Hemisphere winter is consistent with a strengthening of the polar vortex during this period.

**4.5 Wind Speed**

**4.5.1 Stratospheric Winds**

The effect that the aerosol-induced stratospheric temperature perturbation has on the zonal mean windspeed during Northern Hemisphere winter is shown in Fig. 5.





\*\*\*Figure 5\*\*\*


As in Shindell *et al*. (2001, their Plate 5), the left-hand panels in Fig. 5 show that in both UKESM1 and CESM2-WACCM6 a strong stratospheric zonal mean wind anomaly develops at around 10 hPa at 60°-70° N with an increase of more than 12 m s$^{-1}$ for UKESM1 and 9 m s$^{-1}$ for CESM2-WACCM6, thereby enhancing the strength of the polar vortex. The maximum increase in the zonal wind at this level is centred over Alaska in both models (right-hand panels in Fig. 5).

**4.5.2 Tropospheric Winds**

Fig. 5 shows the propagation of this enhanced westerly flow to lower levels in the troposphere and to the surface, with both models suggesting an increased westerly flow north of around 50° N. Fig. 6 shows the Northern Hemisphere wintertime zonal mean wind perturbation at 850 hPa induced by SAI for both models.

\*\*\*Figure 6\*\*\*

As with the stratospheric winds, both models show similar behaviour. Both show enhanced 850 hPa winds particularly over the northern Atlantic between the southern tip of Greenland and the UK. This increased westerly flow penetrates into northern Eurasia indicating that zonal flow is enhanced.

**4.6 Mean Sea Level Pressure and NAO Index**

As noted in section 1, the NAO may be quantified in terms of the pressure difference between Iceland and the Azores. Here we use December-February mean sea-level pressure (MSLP) from the nearest model gridcell to Stykkisholmur, Iceland (65° 05´ N, 22° 44´ W) and Ponta Delgada in the Azores (37° 44´ N, 25° 41´ W). We also construct an NAO index by removing the long-term mean from the timeseries of each location's MSLP, normalising the resulting anomalies by their standard

deviation, and then taking the difference between the normalised anomalies (*e.g.* Hurrell, 1995; Rodwell *et al*., 1999). A positive NAO index indicates when the pressure difference between the two stations is greater than normal and a negative phase when the pressure difference is less than normal. The perturbation to the mean Northern Hemisphere winter surface pressure patterns from SAI is shown in Fig. 7.

\*\*\*Figure 7\*\*\*

Both models show similar large-scale perturbations to MSLP with a vast swath of high pressure anomalies centred over the Atlantic Ocean at around 50° N and to the south of Alaska. The patterns of increased MSLP are broadly similar over Eurasia but are subtly different over the continental USA. A strong area of anomalous low pressure is evident towards the pole in both



models and the strongest pressure gradient anomaly is over the northern Atlantic. This area of strong baroclinicity is associated with the strengthening zonal flow shown in Fig. 6. Over the period 2081-2100, SAI causes the NAO index in UKESM1 to change from -0.36 in G6solar to +0.73 in G6sulfur. This corresponds to the Azores to Iceland pressure difference increasing from 16.4 hPa (G6solar) to 22.3 hPa (G6sulfur) indicating a strengthening of the NAO of around +6 hPa which is significant as the standard error due to natural variability is around 1 hPa. In CESM2-WACCM6, the NAO index increases from -0.34

(G6solar) to +0.77 (G6sulfur), corresponding to a change in pressure difference of 21.3 hPa to 25.9 hPa indicating a strengthening of around 4.5 hPa which is again significant compared with natural variability.

         Before concluding that such impacts on the Northern Hemisphere wintertime NAO are an important difference between end-of-century climates produced by the two different forms of SRM geoengineering, we need to assess if there are any systematic

changes in the NAO over the course of the 21$^{st}$ century in the absence of geoengineering. As noted by Deser *et al.* (2017), some studies project a slight positive shift in the probability distribution of the NAO phase by the end of the 21$^{st}$ century. As G6solar and G6sulfur track the temperature evolution of the SSP2-4.5 scenario, we compare 2081-2100 means from each model's CMIP6 ssp245 simulation with present-day (PD, 2011-2030) means constructed from each model's CMIP6 historical and ssp245 experiments. In UKESM1 the change in Azores to Iceland pressure difference between PD and 2081-2100 in SSP2-

4.5 is 17.6 to 17.7 hPa (NAO index essentially unchanged at +0.19) and in CESM2-WACCM6 the corresponding values are 21.3 to 19.8 hPa (NAO index change -0.26 to -0.63). It is therefore clear that the impact of SAI geoengineering on the Northern Hemisphere wintertime NAO dominates over any effects due to global warming over this period.

**4.7 Regional Mid-latitude Temperature**

We have seen that both models simulate the impact of SAI by inducing a positive phase of the NAO with both models showing

similar patterns of response in stratospheric heating, stratospheric and tropospheric winds and MSLP. We now briefly examine the impact of SAI on near-surface temperatures by looking at the difference between G6sulfur and G6solar during the Northern Hemisphere wintertime with a focus on the continental scale. To put these changes in context, by experimental design the temperature changes in all experiments compared with present day (PD) show the expected warming of climate commensurate with the SSP2-4.5 scenario (annual mean changes from PD to 2081-2100 shown in Fig. 1). The purpose of examining regional

changes in temperature is to emphasize that despite the inter-model similarity of response of many dynamical features associated with the NAO, there are considerable inter-model differences in the resulting regional temperatures in some areas.

     ***Figure 8***

Both models indicate that SAI induces broad-scale patterns of temperature perturbation over Eurasia during Northern Hemisphere winter resembling those associated with a positive phase of the NAO observed subsequent to large tropical volcanic eruptions (Shindell *et al.*, 2004), *i.e.* a warming to the north and a cooling to the south of ~50° N (Fig. 8). Explosive


volcanic eruptions provide a very useful, albeit imperfect, analogue for stratospheric aerosol injection geoengineering (Robock *et al*., 2013). The fact that similar temperature patterns are observed following explosive volcanic eruptions, and that the
proposed mechanisms for impacting the strength of the NAO are identical for volcanic and geoengineering cases, suggests that the inducing of positive phases of the NAO under SAI geoengineering is a relatively robust conclusion.

While there are similarities in the broad-scale hemispheric pattern of temperature perturbations, over continental North America the models suggest rather different regional temperature responses. In UKESM1 the induced positive phase of the
NAO from SAI leads to a warming of the eastern side of the continent as observed (Shindell *et al*., 2004) as well as over the north-western Atlantic, while CESM2-WACCM6 suggests a general cooling across the continent with only the warm anomaly over the North Atlantic being evident. This cooling in CESM2-WACCM6 is consistent with the high-pressure anomaly across the whole continent in this model (Fig. 7) which would enhance advection of cold air from higher latitudes. In contrast, UKESM1 has a low pressure anomaly over much of continental North America which would have the opposite tendency. It is
generally accepted that northern hemispheric wintertime conditions over the eastern USA are anomalously warm during the positive phase of the NAO (*e.g*., http://climate.ncsu.edu/images/edu/NAO2.jpg) which perhaps indicates that UKESM1 may reproduce this phase of the NAO with greater fidelity. In contrast, however, CESM2-WACCM6 seems to better represent the cooling observed at high latitudes over North America following large volcanic eruptions. Significant cooling is also observed over North Africa following such eruptions with cold anomalies extending to around 10° N (Shindell *et al*., 2004). Both models
show cool anomalies in this region but they extend further south in UKESM1 compared with CESM2-WACCM6, suggesting a somewhat weaker response to SAI in the latter model. Reasons for these differences are beyond the scope of this work but demonstrate that important inter-model differences still exist in state-of-the-art climate models.

### 4.8 Regional Mid-latitude Precipitation

Over Europe, while the models exhibit some differences in the exact demarcation between increased precipitation over northern
Europe and Scandinavia and decreased precipitation over southern Europe (Fig. 9), the general patterns are clearly in line with observations during positive phases of the NAO. For example, Fowler and Kilsby (2002) and Burt and Howden (2013) investigated precipitation anomalies in northern areas of the UK and concluded that precipitation and stream-flow is considerably enhanced during positive phases of the NAO. On larger scales, López-Moreno *et al*. (2008) and Casanueva *et al.* (2014) conclude that during the positive phase of the NAO, positive precipitation anomalies occur over northern Europe while
negative precipitation anomalies occur over southern Europe. Furthermore, the study of Zanardo *et al.* (2019) indicates that the NAO clearly correlates with the occurrence of catastrophic floods across Europe and the associated economic losses, and that over northern Europe the majority of historic winter floods occurred during a positive NAO phase.

***Figure 9***






Over North America, both models are consistent and indicate an increase in wintertime precipitation which is again consistent with observations of wintertime precipitation anomalies during the positive phase of the NAO. There are fewer quantitative studies of the impacts of the NAO over North America as the social and economic costs are not so readily apparent as over Europe. However, an analysis by Durkee *et al.* (2008) indicates positive anomalies of rain over south eastern states and positive
anomalies of snowfall over north eastern states during positive phases of the NAO.

### 4.9 Contextualizing in Terms of Changes Compared with Present-day Precipitation

We have shown that the SAI-induced response of the NAO and the associated impacts on precipitation are relatively well understood and reasonably consistent between the two models. As in earlier modelling and observational studies the impact is particularly marked over Europe, with northern Europe experiencing enhanced precipitation and southern Europe reduced
precipitation. We therefore focus our attention on the magnitude of the SAI-induced feedbacks on precipitation from the positive NAO anomaly compared with the temperature-induced feedbacks on precipitation from global warming over the European area. We do this by comparing end of century (2081-2100) precipitation in UKESM1 and CESM2-WACCM6 with that from the present day (PD, 2011-2030) for the ssp585, ssp245, G6solar and G6sulfur simulations (Fig. 10 for UKESM1 and Fig. 11 for CESM2-WACCM6).


***Figure 10***

As expected, Fig. 10 shows that the precipitation changes in 2081-2100 compared with PD are significantly less in ssp245 than in ssp585. North of 50° N there are many areas in ssp585 that experience a change in precipitation exceeding +0.5 mm
day$^{-1}$ while south of 45° N areas tend to be drier than in PD; these patterns are consistent with the patterns of precipitation and runoff changes in multiple-model climate change simulation assessments (Kirtman *et al.*, 2013; Guerreiro, *et al.*, 2018). When comparing the future precipitation response in G6sulfur to that in ssp245, it is evident that the precipitation anomaly pattern from the NAO induced feedback (Fig. 9) acts to reinforce the temperature-induced precipitation feedback. Compared with ssp245, the precipitation anomaly in G6sulfur is more positive in northern Europe and more negative in southern Europe, with
a negative anomaly that encompasses the area all around the Black Sea. When comparing the future precipitation response in G6sulfur with G6solar it is evident that while the precipitation increases north of around 50° N show some consistency between the two, there is no such agreement further south. Over Iberia, Italy, the Balkans, Greece, Turkey, Ukraine and southern Russia the precipitation anomalies show a wintertime precipitation decrease in G6sulfur but an increase in G6solar. It is therefore evident that the idealised approach of G6solar does not adequately represent the regional impacts on precipitation over Europe.


***Figure 11***





Generally, the conclusions from UKESM1 presented in Fig. 10 are supported by the results from CESM2-WACCM6 (Fig. 11). The strong signal of increased precipitation in northern Europe hemisphere evident in ssp585 is reduced in ssp245, G6solar

and G6sulfur. G6sulfur again shows a greater reduction in precipitation south of about 45° N when compared with G6solar. The implications of these findings are discussed in more detail in the following section.

## 5 Discussion and Conclusions

Using data from two Earth system models, we have compared the final 20 years from two numerical experiments which employ different representations of geoengineering in a scenario where the amount of cooling generated is the same. The G6solar

experiment achieves the required cooling by the highly idealised method of reducing the solar constant over the course of the $21^{st}$ century, while the G6sulfur experiment achieves the same degree of cooling by injecting increasing amounts of $SO_2$ into the tropical lower stratosphere (SAI geoengineering). Comparing the results from the two experiments should help cast light on geoengineering impacts which only become evident when the method of geoengineering is represented with some fidelity.

Although both models' SAI simulations are successful in cooling from SSP5-8.5 to SSP2-4.5 levels, the resulting perturbations to the AOD distribution are by no means identical. Differences far larger than these have been reported in earlier coordinated GeoMIP simulations. Pitari *et al*. (2014; their Fig. 3d) indicate that some models (*e.g.* GEOSCCM) perform similarly to UKESM1 in maintaining a peak AOD of three times that at mid-latitudes in the tropical reservoir, while other models (*e.g.* GISS-E2-R) show almost the opposite behaviour with a peak AOD twice that in the tropical reservoir at mid-

latitudes. Pitari *et al*. (2014) caution that aspects of the performance of these two models are hampered by the lack of explicit treatment of heterogeneous chemistry (GISS-E2-R) and the lack of impact of the stratospheric aerosol on photolysis rates (GEOSCCM); these caveats do not apply to the UKESM1 and CESM2-WACCM6 models which include these processes.

The results from both models indicate that a key impact of tropical SAI geoengineering is the generation of a persistent

positive phase of the NAO during Northern Hemisphere wintertime. The intensification of the stratospheric jet produces an increase in surface zonal winds over the North Atlantic leading to a warming of the Eurasian continent northwards of about 50° N and the associated risks of flooding in northern European regions (*e.g.* Scaife *et al*., 2008). The mechanism for generating these anomalies appears to be the same as that observed following large explosive volcanic eruptions in the tropics. This is consistent with the form of SAI simulated in G6sulfur being essentially equivalent to a continuous large volcanic eruption in

the tropics and indicates that the response to any putative continuous large-scale $SO_2$ injection is likely to be the same as that observed for large sporadic eruptions. Unlike some previous findings which suggested that aerosol heating in the lower tropical stratosphere is not necessary to force a positive NAO response (Stenchikov *et al*., 2002), such a response is absent in G6solar in both models considered here. This implicates the warming induced by stratospheric aerosols as a key process in forcing the positive phase of the NAO and associated meteorological impacts as suggested by Shindell *et al*. (2004).






In terms of impacts, the end of century (2081-2100) European wintertime precipitation anomalies in ssp585, ssp245, G6solar and G6sulfur provide an example relating to a critical argument that has been circulating in the geoengineering community for over a decade: that of winners and losers (*e.g*. Irvine *et al*., 2010; Kravitz *et al*. 2014). While few would argue against the benefits of ameliorating the changes in wintertime precipitation under SSP5-8.5 by following the SSP2-4.5 scenario (Figs. 10

and 11), the situation is different when examining the changes seen in G6sulfur. For example, taking the results from CESM2-WACCM6 at face value, one might argue that the impacts of the wintertime drying of vast swathes of the European continent surrounding the Mediterranean Sea (Fig. 11) might be more damaging in terms of their impact on biodiversity, ecology and peoples' lives than the impact of increased flood risk in northern Europe under even the extreme SSP5-8.5 scenario. Of course, here we are limited to analysing the results from just two Earth system models which take no account of trying to tailor the

injection strategy to minimise residual climate impacts (*e.g*. MacMartin *et al*., 2013) and studies have shown that SAI can ameliorate many regional impacts of climate change (*e.g*. Jones *et al*., 2018). Nevertheless, the impact of the SAI-induced effects on the NAO indicate the need for detailed modelling of geoengineering processes when considering the potential regional impacts of such actions. Studies which have investigated the issue of geoengineering winners and loser have generally studied results from idealised solar reduction approaches to geoengineering and therefore may have missed some of the effects

shown here.

In addition to the potential climate impacts from SAI shown here, such intervention would produce many other benefits and risks (*e.g*. Robock, 2020). Some of these additional risks are related not just to the physical climate system, but deal with governance, unknowns, ethics and aesthetics. Furthermore, the technology to inject sulfur into the stratosphere does not

currently exist. Before any decision by society to start climate intervention, much more work is needed to quantify all these potential benefits and risks. In the meantime, even if some climate intervention is used for a time, there remains a great deal of work on mitigation and adaptation to address the threat of global warming.





*Code and data availability*. Due to intellectual property rights restrictions we cannot provide either the source code or documentation papers for the Met Office Unified Model. The UM is available for use under licence - for further information
on how to apply for a licence, see http://www.metoffice.gov.uk/research/modelling-systems/unified-model (last access: 23 July 2020). Previous and current CESM versions are freely available at http://www.cesm.ucar.edu/models/cesm2 (last access: 23 July 2020).

   *Data availability*. The simulation data used in this study are archived on the Earth System Grid Federation (ESGF) (https://esgf-node.llnl.gov/projects/cmip6; last access: 23 July 2020). The model Source IDs are UKESM1-0-LL for UKESM1
and CESM2-WACCM for CESM2-WACCM6.

*Author contributions*.  AJ and JMH led the analysis and wrote the manuscript with contributions from ACJ, ST, BK and AR.
The UKESM1 and CESM2-WACCM6 simulations were carried out by AJ and ST, respectively. BK was central in co-ordinating the GeoMIP6 activity.

*Competing interests*. The authors declare that they have no competing interests.

*Acknowledgements*   AJ and JMH were supported by the Met Office Hadley Centre Climate Programme funded by BEIS and
Defra. AJ would like to thank the Met Office team responsible for the *managecmip* software which greatly simplified the work involved. The CESM project is supported primarily by the National Science Foundation (NSF). Some of the material is based upon work supported by the National Center for Atmospheric Research (NCAR), which is a major facility sponsored by the NSF under Cooperative Agreement No. 1852977. Computing and data storage resources for CESM, including the Cheyenne supercomputer (doi:10.5065/D6RX99HX), were provided by the Computational and Information Systems Laboratory (CISL)
at NCAR. Support for BK was provided in part by the NSF through agreement CBET-1931641, the Indiana University Environmental Resilience Institute, and the *Prepared for Environmental Change* Grand Challenge initiative. The Pacific Northwest National Laboratory is operated for the US Department of Energy by Battelle Memorial Institute under contract DE-AC05-76RL01830. AR is supported by NSF grant AGS-2017113. We acknowledge the World Climate Research Programme which, through its Working Group on Coupled Modelling, coordinated and promoted CMIP. We thank the climate
modelling groups for producing and making available their model output, ESGF for archiving the data and providing access,



and the multiple funding agencies who support CMIP6 and ESGF. We also thank all participants of the Geoengineering Model Intercomparison Project and their model development teams.

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




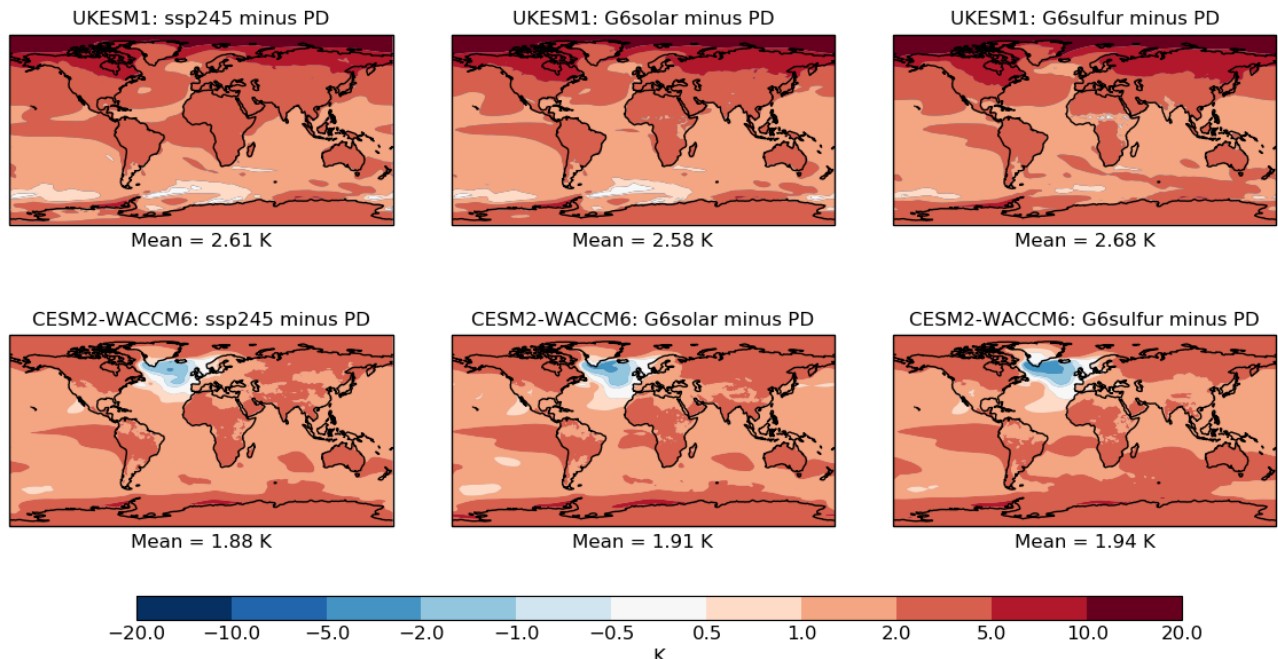

**Figure 1: Annual mean temperature change (K) from present day (PD; 2011-2030 mean) to the end of the century (2081-2100 mean) in the various experiments. Upper row shows results from UKESM1, lower row for CESM2-WACCM6. PD data are taken from years 2011-2014 of each model's CMIP6 historical simulation combined with years 2015-2030 from the corresponding ssp245 simulation. All results are ensemble means (three members for UKESM1, two for CESM2-WACCM6).**





**Figure 2: The distribution of the 2081-2100 mean anomaly in annual mean AOD at 550nm (dimensionless) due to stratospheric SO₂** injection for UKESM1 (upper left), CESM-WACCM6 (lower left) and zonal means for both models (right). The anomaly is calculated from the difference between G6sulfur and G6solar.





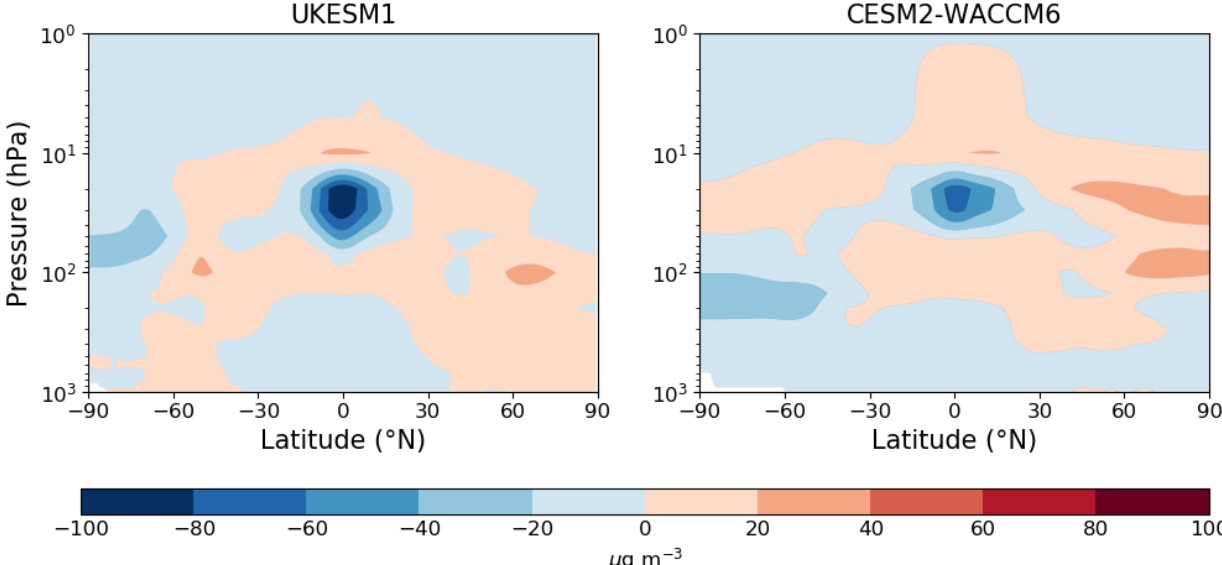

**Figure 3: The difference in 2081-2100 annual mean ozone concentrations (µg m⁻³) diagnosed from {G6sulfur minus G6solar} for UKESM1 (left) and CESM2-WACCM6 (right).**


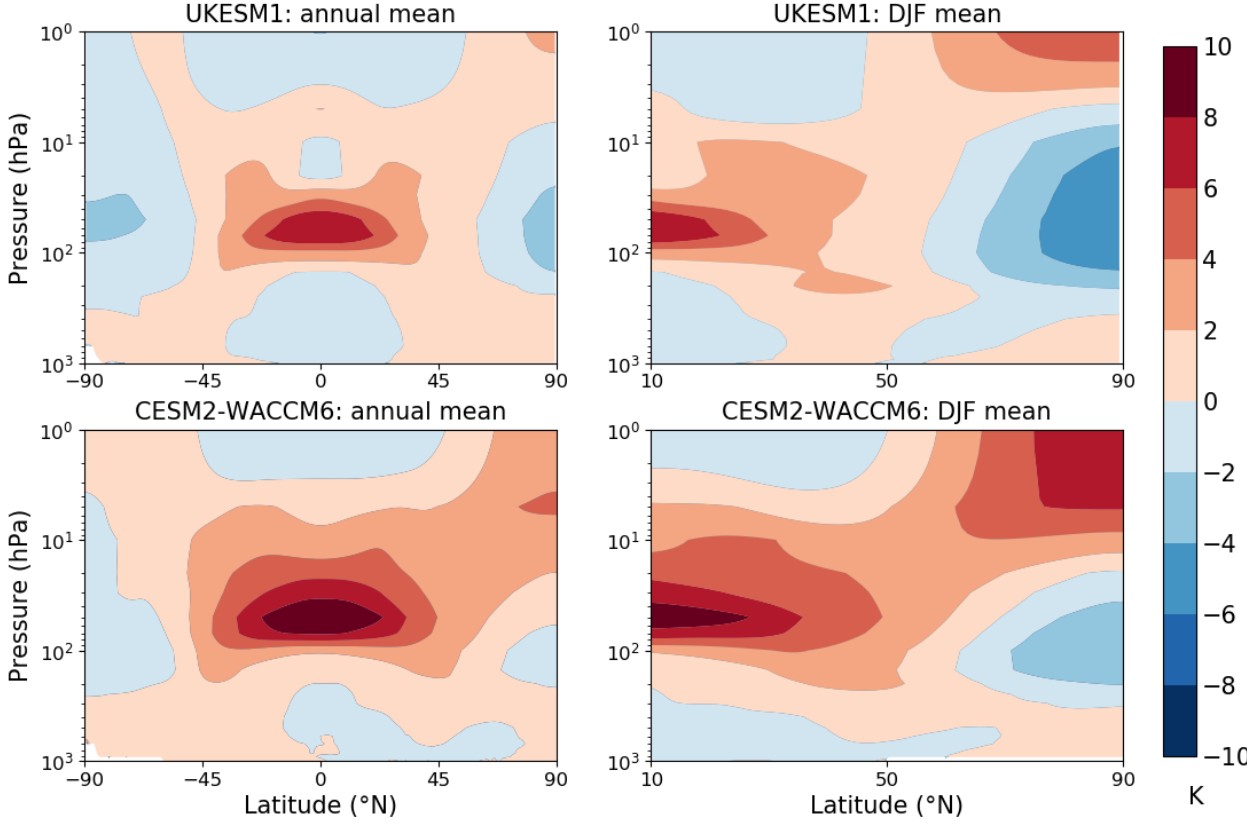

**Figure 4: The difference in zonal mean temperature (K) diagnosed from {G6sulfur minus G6solar}; the upper panels show results from UKESM1 and the lower from CESM2-WACCM6. The panels on the left show global annual-mean results from 2081-2100, those on the right show Northern Hemisphere winter (December-February) means over the same period.**





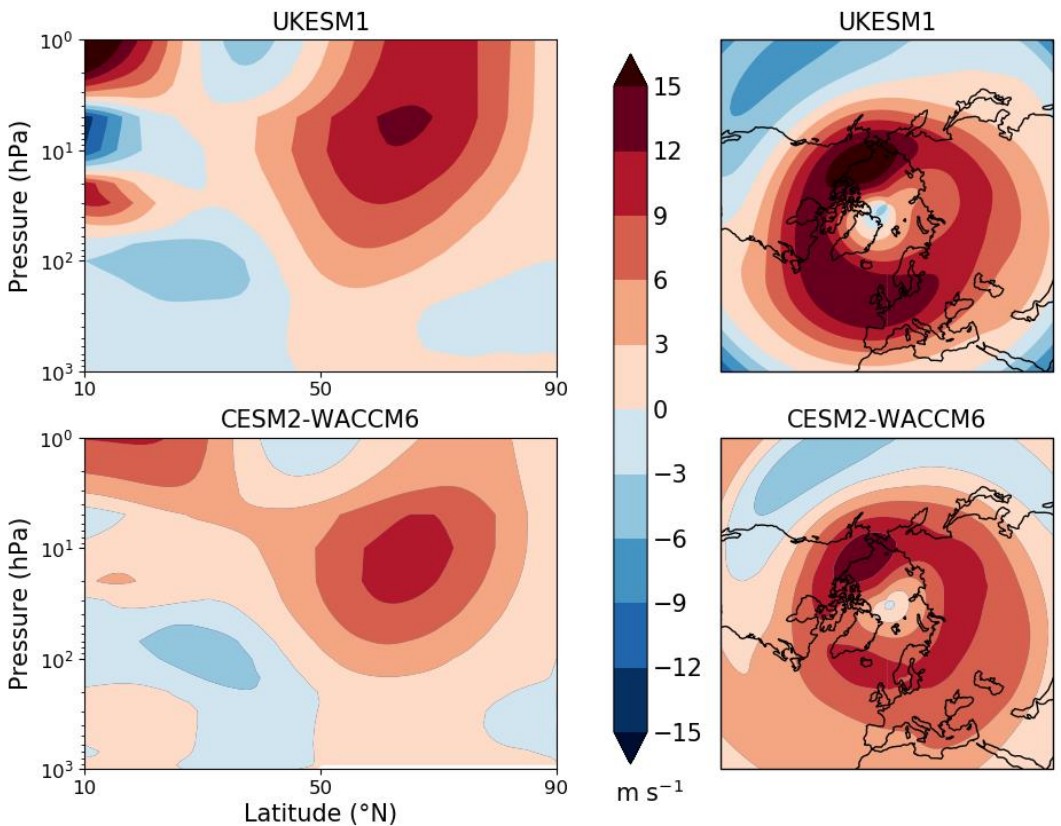

**Figure 5: The perturbation to mean December-February zonal wind speed over 2081-2100 (m s⁻¹) caused by SAI, diagnosed from {G6sulfur minus G6solar}. The left-hand panels show the change in Northern Hemisphere zonal wind, positive values indicating a westerly perturbation and negative values an easterly one. The right-hand panels show the spatial distribution of this change at 10 hPa, the level of maximum perturbation. The upper panels show results from UKESM1, the lower from CESM2-WACCM6.**






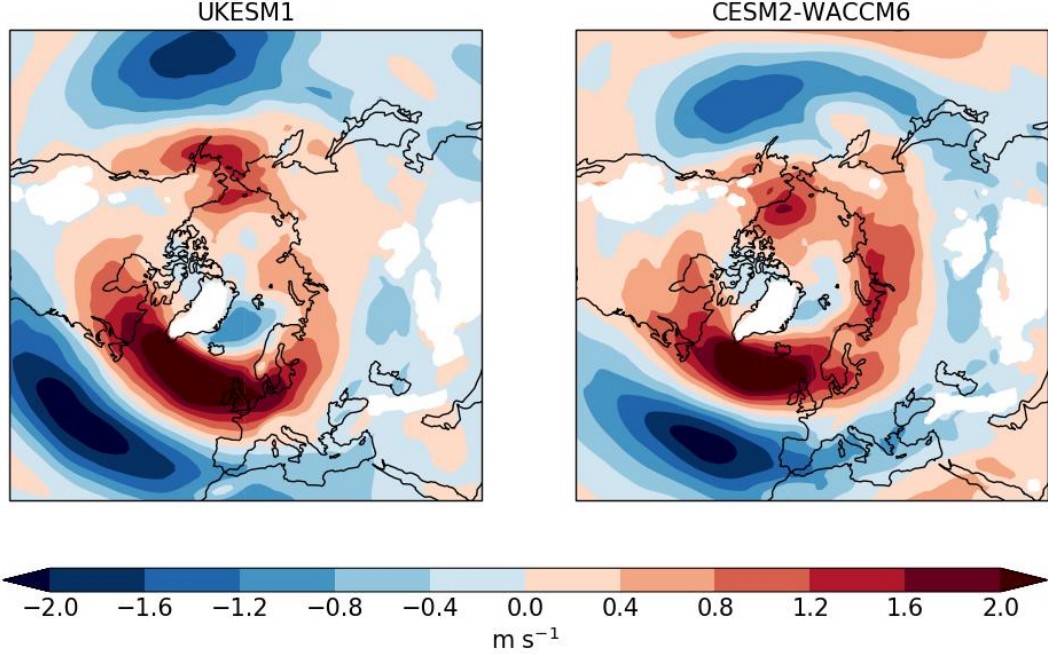

**Figure 6: The distribution of the 2081-2100 mean December-February zonal wind speed perturbation due to SAI at 850hPa (m s⁻¹) for UKESM1 (left) and CESM2-WACCM6 (right). Positive values represent a westerly perturbation and negative values an easterly perturbation; white areas indicate regions where the surface elevation is higher than the mean 850 hPa pressure level.**






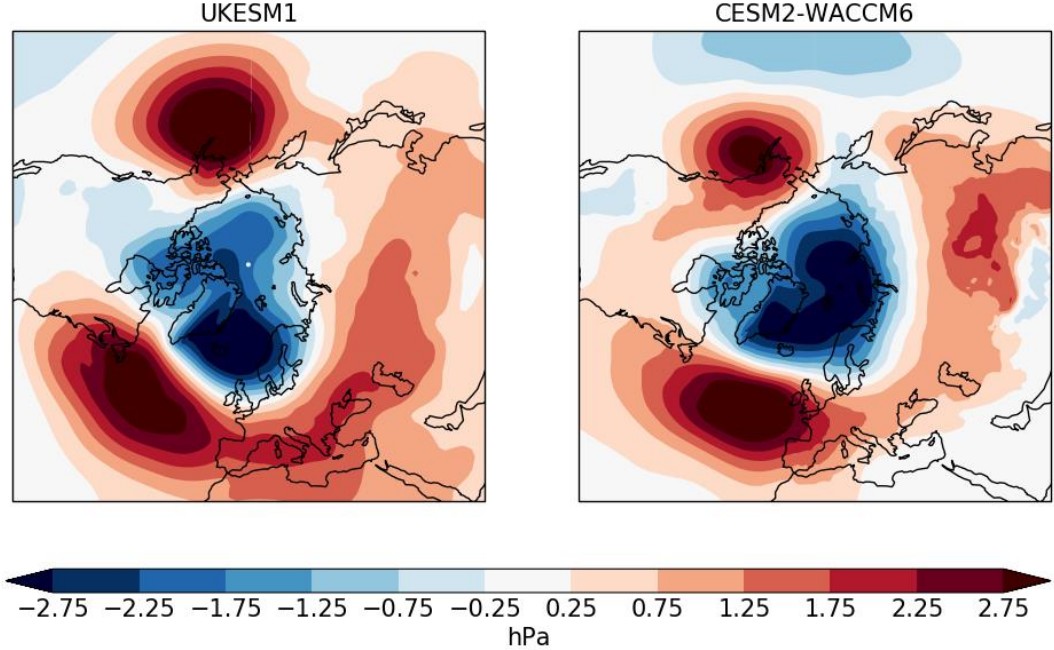

**Figure 7: The change induced by SAI in 2081-2100 mean December-February MSLP (hPa) for UKESM1 (left) and CESM2-WACCM6 (right) diagnosed from {G6sulfur minus G6solar}.**


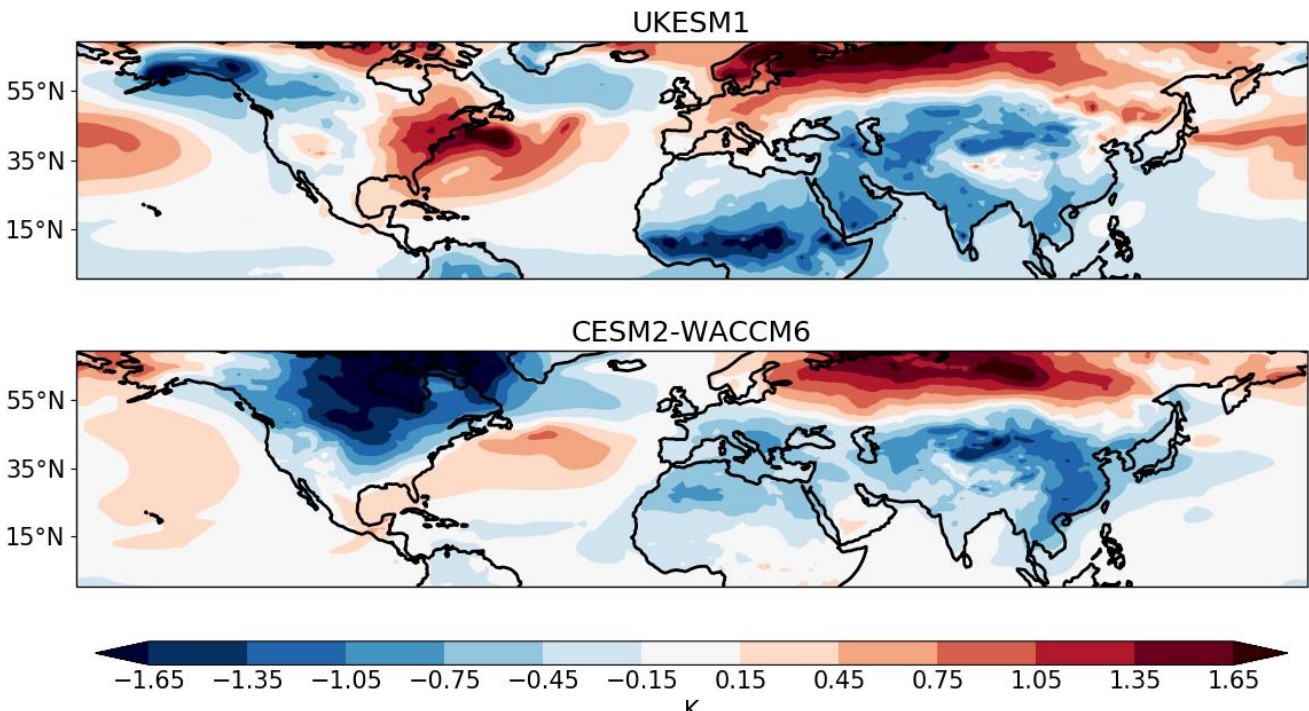

**Figure 8: The perturbation to 2081-2100 mean December-February near-surface air temperature (K) induced by SAI diagnosed from {G6sulfur minus G6solar} for UKESM1 (upper panel) and CESM2-WACCM6 (lower panel). The area plotted is chosen to replicate that presented by Shindell *et al.,* (2004), their Fig. 2.**





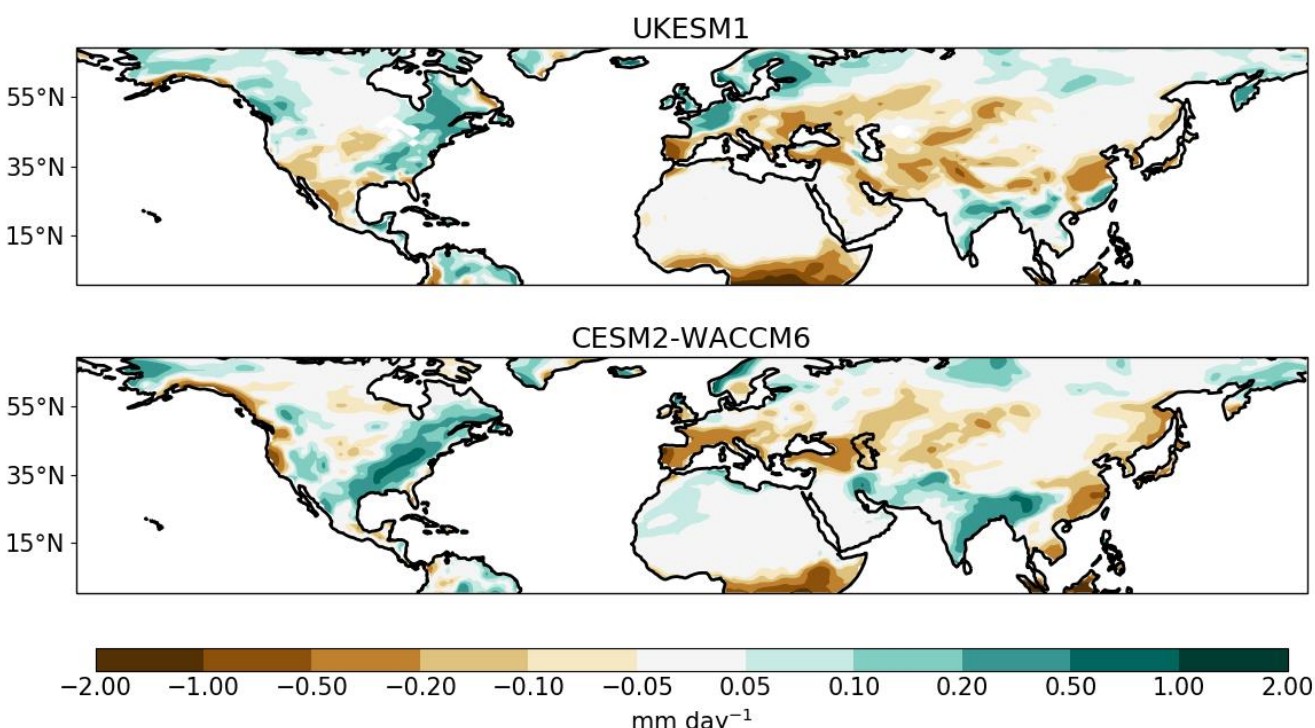

**Figure 9: The perturbation to 2081-2100 mean December-February land precipitation rate (mm day[-1]) induced by SAI diagnosed from {G6sulfur minus G6solar} for UKESM1 (upper panel) and CESM2-WACCM6 (lower panel).**






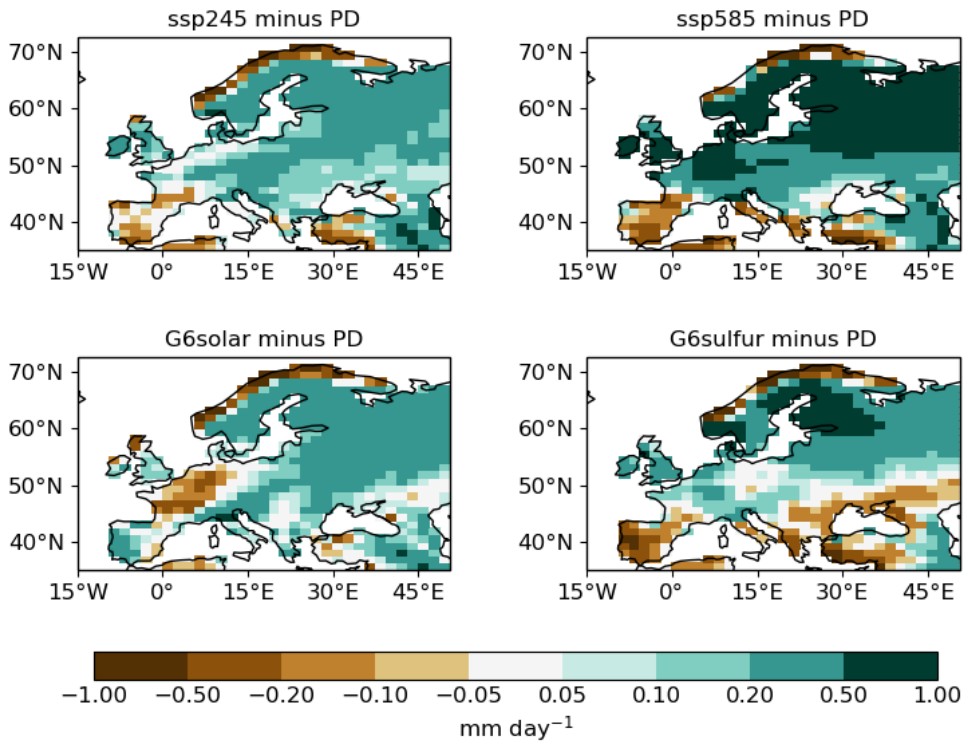


**Figure 10: Changes in mean December-February land precipitation rate (mm day⁻¹) between present day (PD, 2011-2030) and 2081-2100 in experiments ssp245, ssp585, G6solar and G6sulfur in UKESM1. PD means are constructed in the same manner as in Fig. 1.**








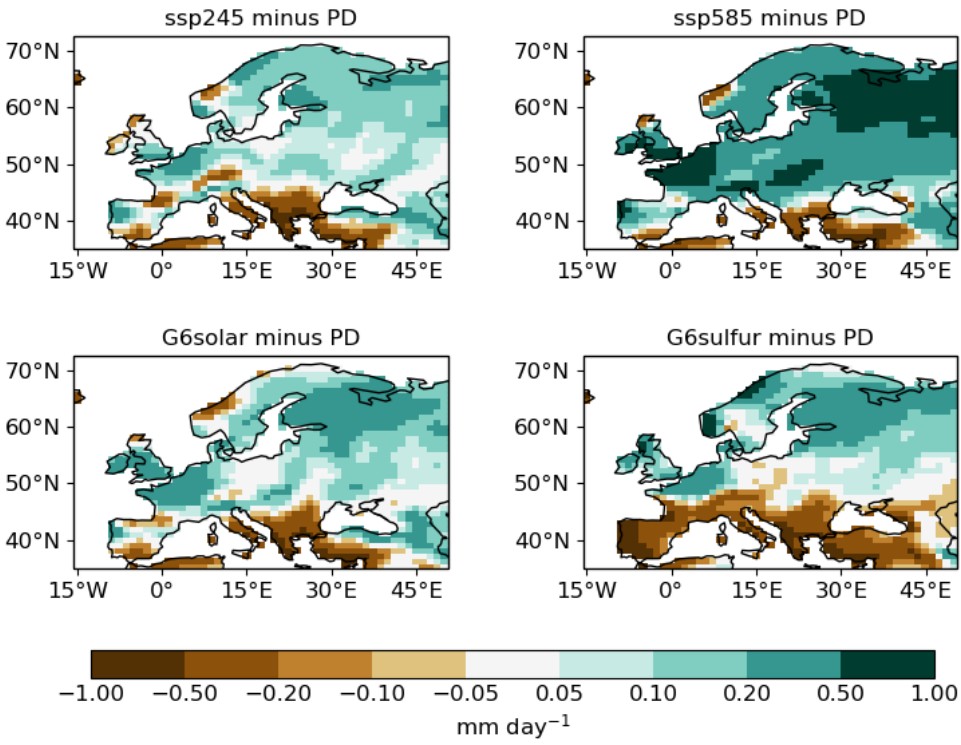

Figure 11: As Fig. 10 but for CESM2-WACCM6.