# Peer review of "North Atlantic Oscillation response in GeoMIP experiments G6solar and G6sulfur: why detailed modelling is needed for understanding regional implications of solar radiation management"

_Atmospheric Chemistry and Physics, 2020_

## Short Comment (SC1) · 17 Aug 2020

Before analyzing GHG influence on NAO, need to analyze the natural forcing response. See attached figure and the study it is based on.
* * *
[Figure]

[Figure]

**Fig. 1.**

---

## Author Comment (AC1) · 25 Aug 2020

Thank you for your comment. Our study is not investigating any changes to the NAO due to natural or GHG forcing. We are only looking at the differences between G6sulfur and G6solar.

---

## Referee Comment (RC1) · Anonymous Referee #1 · 5 Sep 2020

General concerns:

This study is based on the analysis of the GEOMIP simulations of two prominent models from those participating in the MIP, UKESM1, and WACCM6. The authors compare G6solar and G6sulfur experiments from GEOMIP. These experiments are devoted to reducing warming in the SSP5-8.5 IPCC scenario to the SSP2-4.5 scenario, decreasing solar constant (G6solar), or injecting SO2 in the lower stratosphere (G6sukfur). The solar or sulfate aerosol forcings are not alined in different experiments and two models, but calibration is based on the global surface air temperature. The title of the

paper is misleading. The NAO response in G6sulfur is the most exciting result, but it is not only about this. The paper is well written and logically organized. Still, the authors fail to put their findings, at least their NAO-response results, in a context of a few decades-long research of NAO/AO sensitivity to solar and volcanic forcings. Suppose the authors search using names of Hans Graf or Kunichiko Kodera. In that case, they will find plenty of publications with a wealth of information that is closely related to what is discussed in the current paper.

The authors mention several times that SAI has a significant advantage of other geo-engineering techniques because it has an "imperfect" but useful natural analog - volcanic eruptions. But unfortunately, they never use this. E.g., it was never asked if the equator-pole temperature gradient in the lower stratosphere calculated within UKESM1 and WACCAM-6 is realistic. It is known that. e.g., WACCAM overestimates stratospheric temperature response to volcanic eruptions.

Finally, the authors should more clearly formulate their study's objective and what they want to achieve.

Specific comments:

L72: Graf and Kodera discussed this phenomenon much earlier.

L74: It is an incorrect interpretation of the point stated in (Polvani et al., 2019). They reject a casual link between volcanic forcing and AO's positive phase following the 1991 Pinatubo eruption. Stenchikov et al. (2006) and Driscoll et al. (2012) discussed the signal's low amplitude in the existing models.

L85-90: Please formulate the objectives.

L165: UKESM1 and WACCAM6 have absolutely different climates by the end of the 21st century. It deserves a little more explanation.

L168-172: Is it your objective? Why is it here?

L174-176: This is Pinatubo-size emission annually, a colossal forcing.

L202-205: The equatorial lower stratosphere is overheated by 10K. Was it realistically calculated? Could you put this in the context of model results for the 1991 Pinatubo eruption?

L213-214: This is the incorrect statement. Stenchikov et al. (1998) attributed 1/3 of stratospheric heating to solar radiation absorption by sulfate aerosols near IR and 70% - IR absorption.

L216-217: It is not only a lack of solar radiation but also a low IR flux because of low temperatures.

Section 4.7: The first simulations of volcanic impact on climate were conducted by reducing solar constant (see Soden et al. 2002). So the difference between SO2 injection and changing of the solar constant was known. The reduction of the solar constant, by the way, should cause a negative NAO response. This will add in the G6sulfur-G6solar signal. Another point is that G6sulfur produces an extreme temperature meridional gradient in the lower stratosphere, five times made by the 1991 Pinatubo eruption. This is the reason why we have this stable winter warming response. Weather models calculated this response correctly remains open until it is tested in observations.

L307: Strange to use this argument here, when UKESM1 and WACCAM6 produce absolutely different climates with strongly different AMOC intensity.

L359: It is not only the amount of cooling that matters but a change in circulation

L378: It was not observed but suggested. The question is still open.

L381-384: This is a misinterpretation. Stenchikov et al. (2002) stated that along with the stratospheric mechanism (suggested by Graf and Kodera), the polar cooling due to polar ozone depletion and tropospheric planetary wave response could contribute in the positive NAO response.

[Figure]

There is a vast literature on NAO response to solar forcing (see Kodera's papers). It would be useful to compare the G6solar responses with that results. That would be a test for the models. There is no discrepancy (as for volcanic sulfate aerosols) between solar geoengineering forcing and natural forcing. Because in both cases, solar forcing is stationary.

---

## Referee Comment (RC2) · Anonymous Referee #2 · 7 Sep 2020

General comments: This paper uses data from GEOMIP, where they compare two different solar radiation management techniques. One is a decreasing solar constant (G6Solar) and one is injection so2 in the stratosphere(G&Sulfur). In these experiments warming in the SSP5-8.5 scenario is reduce to SSP2-4.5. The author compares results from two climate models UKESM1 and WACCM6. The tittle of the paper is "North Atlantic Oscillationresponse in GeoMIP experiments G6solar andG6sulfur: why detailed modelling is needed for understanding regional implications of solar radiation management", however the main results are not clearly connecting NAO to the observed

changes. Main results are temperature, precipitation, and mean sea surface pressure response to different SRM techniques, and author does not show clearly how these responses are depending on the phase of NAO. This paper highlight the importance of atmospheric dynamical response to different SRM techniques and especially aerosols dynamical resposen.

Selected model are quite different from each other, UKESM1 goes up to 85km where WACCM6 goes to 140km. How this affects to the results? Also author should include what aerosol-cloud proses are included in these models.

This manuscript miss clear definition of NAO, Author should include the formula that they used to calculate NAO. Also author refer to different phase of NAO in the text, example in line 314. I recommend to included figures where the responses i.e for precipitation is shown separately for NAO positive and negative phase.

In the result sections line 146 definition of present day run is not clear, it has been stated that PD is mean of 2011 - 2030, however what ssp scenario is used here is unclear.

For reader it would be helpfull if all results are also showed respect to the present day

Specific comments:

Line 23 :In Abstract author should include model names

Line 25: In abstract when author refers regional warming, spesifiy whitch regions.

Line 26: "These findings are broadly consistent with previous findings on the impact of stratospheric volcanic aerosol on the NAO" specify this. What are the previous findings

Line 36: author talks about aerosol-cloud interactions, author should specify the different interactions mechanisms.

Line 70: This sections deals of definitions of NAO. This should be in Method section.

Line 85-90: Deals with model selections, this should also be in method sections and include some arguments are the model independent

Line 144: Define key variables

Line 151: Include the difference picture

Line 185: Include more

---

## Author Response (AR1)

Response to Reviewer #1

We would like to thank the Reviewer for their comments. There were a number of useful constructive criticisms that we have
included in our revised manuscript and we address each of these in turn below. The Reviewer's comments are reproduced in
**bold** throughout.

**General concerns**

**This study is based on the analysis of the GEOMIP simulations of two prominent models from those participating in
the MIP, UKESM1, and WACCM6. The authors compare G6solar and G6sulfur experiments from GEOMIP. These
experiments are devoted to reducing warming in the SSP5-8.5 IPCC scenario to the SSP2-4.5 scenario, decreasing solar
constant (G6solar), or injecting SO2 in the lower stratosphere (G6sukfur). The solar or sulfate aerosol forcings are not
alined in different experiments and two models, but calibration is based on the global surface air temperature. The title
of the paper is misleading. The NAO response in G6sulfur is the most exciting result, but it is not only about this. The
paper is well written and logically organized. Still, the authors fail to put their findings, at least their NAO-response
results, in a context of a few decades-long research of NAO/AO sensitivity to solar and volcanic forcings. Suppose the
authors search using names of Hans Graf or Kunichiko Kodera. In that case, they will find plenty of publications with
a wealth of information that is closely related to what is discussed in the current paper.**

**The authors mention several times that SAI has a significant advantage of other geo-engineering techniques because it
has an "imperfect" but useful natural analog - volcanic eruptions. But unfortunately, they never use this. E.g., it was
never asked if the equator-pole temperature gradient in the lower stratosphere calculated within UKESM1 and
WACCAM-6 is realistic. It is known that. e.g., WACCAM overestimates stratospheric temperature response to
volcanic eruptions.**

**Finally, the authors should more clearly formulate their study's objective and what they want to achieve.**

The Reviewer states that "**The title of the paper is misleading**". We believe that the title adequately summarises what we are
trying to address. Note that we have responded to the Reviewer's comment that we should "**more clearly formulate their
study's objective and what they want to achieve**" below. This now links the aims and objectives of the research more clearly
to the title of the paper.

The Reviewer states that the paper is "**well written and logically organized**", which is gratifying. The main criticism is that
links to previous work in terms of NAO response to volcanic and solar forcings are not well enough documented in the present
paper. We believe that we do generally include sufficient background on volcanic forcing and the NAO. In the introductory
paragraph on the NAO we include references to the following papers, all of which focus on the dynamical response subsequent to injections of $SO_2$ from volcanic eruptions into the stratosphere: Robock and Mao (1992); Hurrell (1995); Stenchikov *et al*. (2002); Lorenz and Hartmann (2003); Shindell *et al*. (2004), Polvani *et al*. (2019).

The reason for the lack of discussion regarding the dynamical and NAO response to solar variability is that the paper is focussed on solar radiation management (SRM) geoengineering as modelled by the GeoMIP project (Kravitz *et al*., 2015). SRM via stratospheric aerosol injection (SAI) is seen as being one of the few plausible mechanisms that could be implemented in any practical deployment to combat global warming. Deploying "mirrors in space" to reflect sunlight (effectively a reduction in the solar constant) is not considered plausible. This is summarised by the Royal Society (2009) report on geoengineering that we reference in our paper (see summary figure below). Space mirrors are simply not affordable at the scales need and would take too long to develop:

[Figure]

Successive reports on geoengineering (*e.g*., National Research Council, 2015; Lawrence *et al*., 2018) have confirmed this viewpoint and hence SAI has become the mainstay of cutting-edge SRM research. Our approach is therefore focussed on the more practical (G6sulfur) rather than the impractical (G6solar).

However, we agree with the Reviewer that we should include some discussion of the work on the relationship between solar variability (particularly the 11-year solar cycle) and the NAO. We therefore include the following paragraphs in the

Introduction (starting at line 86) and now include approximately the same number of references to solar forcings as to SAI, which we believe provides a better balance:

"In addition to work on the dynamical and NAO response to SAI via volcanic eruptions, there has been much debate on the influence of the 11-year solar cycle with stronger solar activity being associated with a positive phase of the NAO and weaker solar activity being associated with a negative phase. Early work (*e.g.* Kodera, 2002; Kodera and Kuroda; 2005; Matthes *et*
*al.*, 2006) suggested that mechanisms influencing the NAO from solar variability originated near the stratopause and
propagated downward through the stratosphere and influence the troposphere via changes in meridional propagation of
planetary waves. More recent work has suggested that stronger correlations exist between the solar cycle and the phase and
strength of the NAO if a lag is accounted for (Gray *et al.*, 2013) owing to ocean-atmosphere interactions that strengthen the
response (Scaife *et al.*, 2013). These lagged responses to solar cycles have been replicated in some climate models (*e.g.* Ineson
*et al.*, 2011), including a version of the model that was the forerunner of the UKESM1 model that is used in our analysis (see
section 2).

Stratospheric aerosol and the 11-year solar cycle are not the only phenomena to influence the NAO: Smith *et al.* (2016)
indicated that Atlantic sea surface temperatures, the phase and strength of El Niño, the quasi-biennial oscillation, Atlantic
multi-decadal variability, and Pacific decadal variability may all play a role. However, skilful predictions of the wintertime
NAO index using sophisticated seasonal prediction models that account for these factors are now possible (Dunstone *et al.*,
2016). The two driving mechanisms investigated in this study, SAI and a reduction in solar constant, may induce opposing
impacts on the NAO: SAI might strengthen the NAO, while reducing the solar constant might weaken it."

To respond to the Reviewer's criticism and better state the aims and objectives of the paper we include the following in the
penultimate paragraph of the Introduction (line 111 onwards):

"The main objective is to determine whether, under SRM strategies which are continuous rather than sporadic or periodic in
nature, the two models produce NAO responses that are consistent with the expectations discussed above: that SAI induces a
significant shift to the positive phase of the NAO compared with reducing the solar constant. Our analysis focuses on the
broad-scale microphysical, chemical and dynamical features in the Northern Hemisphere winter, *i.e.* aerosol spatial
distributions, impacts on ozone, stratospheric temperatures, stratospheric and tropospheric zonal mean winds and induced
surface pressure patterns with a focus on the NAO, before examining impacts on continental-scale temperature and
precipitation patterns. SAI is considered the most plausible SRM method owing to considerations of effectiveness, timeliness,
cost and safety (*e.g.* Royal Society, 2009). Our focus is therefore on the difference between the responses to SRM via SAI and
that via generic reductions in the solar constant, noting that many previous assessments of the impacts of SRM use a reduction
of the solar constant as a proxy for SAI."

Dunstone, N., Smith, D., Scaife, A., Hermanson, L., Eade, R., Robinson, N., Andrews, M. and Knight, J., 2016. Skilful
predictions of the winter North Atlantic Oscillation one year ahead. *Nature Geoscience*, **9**, 809-814, doi:10.1038/NGEO2824.
Gray, L. J., Scaife, A. A., Mitchell, D. M., Osprey, S., Ineson, S., Hardiman, S., Butchart, N., Knight, J. R., Sutton, R., and
Kodera, K., 2013. A lagged response to the 11 year solar cycle in observed winter Atlantic/European weather patterns. *J.
Geophys. Res.*, **118**, 13405–13420, doi:10.1002/2013JD020062.

Ineson, S., Scaife, A. A., Knight, J. R., Manners, J. C., Dunstone, N. J., Gray, L. J., and Haigh, J. D., 2011. Solar forcing of winter climate variability in the Northern Hemisphere. *Nature Geosci.*, **4**, 753–757, doi:10.1038/ngeo1282.

Kodera, K., 2002. Solar cycle modulation of the North Atlantic Oscillation: Implication in the spatial structure of the NAO. *Geophys. Res. Lett.*, **29**, 59-1-59-4, doi:10.1029/2001GL014557.

Kodera, K., and Kuroda, Y., 2005. A possible mechanism of solar modulation of the spatial structure of the North Atlantic Oscillation. *J. Geophys. Res.*, **110**, D02111, doi:10.1029/2004JD005258.

Kravitz, B., Robock, A., Tilmes, S., Boucher, O., English, J. M., Irvine, P. J., Jones, A., Lawrence, M. G., MacCracken, M., Muri, H., Moore, J. C., Niemeier, U., Phipps, S. J., Sillmann, J., Storelvmo, T., Wang, H., and Watanabe, S., 2015. The Geoengineering Model Intercomparison Project Phase 6 (GeoMIP6): simulation design and preliminary results, *Geosci. Model. Dev.*, **8**, 3379–3392, doi:10.5194/gmd-8-3379-2015.

Lawrence, M. G., Schäfer, S., Muri, H., Scott, V., Oschlies, A., Vaughan, N. E., Boucher, O., Schmidt, H., Haywood, J., and Scheffran J., 2018. Evaluating climate geoengineering proposals in the context of the Paris Agreement temperature goals, *Nature Communications*, **9**:3734, doi:10.1038/s41467-018-05938-3

Matthes, K., Kuroda, Y., Kodera, K., and Langematz, U., 2006. Transfer of the solar signal from the stratosphere to the troposphere: Northern winter. *J. Geophys. Res.*, **111**, D06108, doi:10.1029/2005JD006283.

National Research Council, 2015. *Climate Intervention: Reflecting Sunlight to Cool Earth*. The National Academies Press, Washington, DC, doi:10.17226/18988.

Royal Society, 2009. *Geoengineering the climate: Science, governance and uncertainty*, RS Policy Document 10/09 RS1636, The Royal Society, London, UK.

Scaife, A. A., Ineson, S., Knight, J. R., Gray, L., Kodera, K., Smith, D. M., 2013. A mechanism for lagged North Atlantic climate response to solar variability. *Geophys. Res. Lett.*, **40**, 434–439, doi:10.1002/grl.50099.

Smith, D. M., Scaife, A. A., Eade, R., and Knight, J. R., 2016. Seasonal to decadal prediction of the winter North Atlantic Oscillation: emerging capability and future prospects. *Quart. J. Royal Meteorol. Soc.*, **142**, 611-617, doi:10.1002/qj.2479.

**Specific comments**

**L72: Graf and Kodera discussed this phenomenon much earlier.**

We now refer to the earlier work of Graf *et al.* (1994) and Kodera (1994) so the text now reads (lines 71-74):

"Both model simulations (*e.g.* Stenchikov *et al.*, 2002) and observations (*e.g.* Graf *et al.*, 1994; Kodera, 1994; Lorenz and Hartmann, 2003) have shown that one of the most significant atmospheric responses following explosive volcanic eruptions is a strengthening of the polar vortex and an impact on the Northern Hemisphere wintertime NAO…"

We also now refer to the earlier work of Graf & Walter (2005; lines 270-271):

"…and shows a strong similarity to the pattern of wind speed perturbation identified in reanalysis data when the polar vortex is strong (*e.g.* Graf and Walter, 2005)."

**L74: It is an incorrect interpretation of the point stated in (Polvani et al., 2019). They reject a casual link between volcanic forcing and AO's positive phase following the 1991 Pinatubo eruption. Stenchikov et al. (2006) and Driscoll et al. (2012) discussed the signal's low amplitude in the existing models.**

We have rephrased this sentence as follows (lines 72-75):

"….have shown that one of the most significant atmospheric responses following explosive volcanic eruptions is the impact on the Northern Hemisphere wintertime NAO, although in the case of the 1991 Pinatubo eruption the causal link has recently been questioned by Polvani *et al.* (2019)."

**L85-90: Please formulate the objectives.**

We are now much clearer about the objectives of the paper in new text we have added towards the end of the Introduction (lines 111-120):

"The main objectives of the research are to determine whether, under SRM strategies which are continuous rather than sporadic or periodic in nature, the two models produce NAO responses that are consistent with the expectations discussed above: that SAI induces a significant shift to the positive phase of the NAO compared with reducing the solar constant. Our analysis focuses on the broad-scale microphysical, chemical and dynamical features in the northern hemisphere winter, *i.e.* aerosol spatial distributions, impacts on ozone, stratospheric temperatures, stratospheric and tropospheric zonal mean winds and induced surface pressure patterns with a focus on the NAO, before examining impacts on continental-scale temperature and precipitation patterns. SAI is considered the most plausible SRM method owing to considerations of effectiveness, timeliness, cost and safety (*e.g.* Royal Society, 2009). Our focus is therefore on the difference between the responses to SRM via SAI and that via generic reductions in the solar constant, noting that many previous assessments of the impacts of SRM use a reduction of the solar constant as a proxy for SAI."

**L165: UKESM1 and WACCAM6 have absolutely different climates by the end of the 21st century. It deserves a little more explanation.**

We disagree that the models have "absolutely different climates" by the end of the 21st century. As Fig.1 shows, both models have warmed considerably under the SSP2-4.5 scenario with the greatest warming over land and at high latitudes. The fact that the amount of warming and its distribution differ between the two models (obviously including the region of cooling in the North Atlantic in CESM2-WACCM6) is what might be expected from two different climate models with different formulations for the various climate components. This paper is not about different model climate sensitivities to global warming *per se* – rather the similarity of the responses to SAI as becomes apparent in the results.

**L168-172: Is it your objective? Why is it here?**

We now explicitly define our objectives in line 111-120 (see above).

**L174-176: This is Pinatubo-size emission annually, a colossal forcing.**

Indeed it is, but that is what is required in the models to achieve the temperature goals required of this GeoMIP experiment. GeoMIP is a CMIP6-endorsed effort and we have to stick to the protocols.

**L202-205: The equatorial lower stratosphere is overheated by 10K. Was it realistically calculated? Could you put this in the context of model results for the 1991 Pinatubo eruption?**

The focus of the paper is not on Pinatubo, it is on SRM geoengineering in the context of the GeoMIP framework. We point out that volcanic eruptions such as Pinatubo provide useful, but not perfect, analogues for SAI and cite suitable literature (*e.g*. Robock *et al*., 2013; line 288 of the original manuscript, lines 320-321 of the revised version). We also state explicitly that the models (or their immediate forebears) have undergone extensive validation against explosive eruptions such as Pinatubo (line 100 of the original submission, now lines 126-128):

"Both UKESM1 and CESM2-WACCM6 are fully coupled Earth system models which have contributed to CMIP6 and GeoMIP6. Both models (or their immediate forebears) have undergone various degrees of validation relevant to SAI using observations from explosive volcanic eruptions (*e.g*. Haywood *et al*., 2011; Dhomse *et al*., 2014; Mills *et al*., 2016)."

It is not reasonable to expect models that are participating in model intercomparison studies such as GeoMIP to first run simulations where volcanic simulations are simulated and the results assessed. All science is built on previous research and by
referencing this prior research we have provided the evidence that the models have been assessed against volcanic eruptions. For the Reviewer's information, VolMIP simulations are being undertaken using UKESM1 but the results are not yet available.

**L213-214: This is the incorrect statement. Stenchikov et al. (1998) attributed 1/3 of stratospheric heating to solar radiation absorption by sulfate aerosols near IR and 70%- IR absorption.**

We thank the Reviewer for pointing this out and have revised the text as follows (lines 241-243):

"…the small amount of absorption of solar radiation by stratospheric aerosols in the near-infrared, together with absorption of terrestrial longwave radiation, cause the stratospheric heating…"

**L216-217: It is not only a lack of solar radiation but also a low IR flux because of low temperatures.**

We agree and have rewritten the text as follows (lines 245-247):

"The right-hand panels of Fig. 4 show that the impact of solar absorption in the stratosphere cannot be effective during the polar night. This, along with a reduced flux of terrestrial radiation due to low wintertime temperatures, means that stratospheric heating from the aerosol is only present at latitudes south of the Arctic Circle"

**Section 4.7: The first simulations of volcanic impact on climate were conducted by reducing solar constant (see Soden et al. 2002). So the difference between SO2 injection and changing of the solar constant was known. The reduction of the solar constant, by the way, should cause a negative NAO response. This will add in the G6sulfur-G6solar signal. Another point is that G6sulfur produces an extreme temperature meridional gradient in the lower stratosphere, five times made by the 1991 Pinatubo eruption. This is the reason why we have this stable winter warming response.**

**Weather models calculated this response correctly remains open until it is tested in observations.**

Soden *et al*. (2002) used a method intermediate between SAI and reducing the solar constant. They report that they used spectrally- and zonally-varying aerosol optical depths to simulate the forcing from the Pinatubo eruption. This is both more complex than a simple reduction to the solar constant and less complex than SAI as they simply prescribed AOD. This does not appear to support the Reviewer's assertion that "**the difference between SO2 injection and changing of the solar**

**constant was known**". We now note in the Introduction (see above) that reducing the solar constant tends to have the opposite effect (in terms of the NAO response) to that caused by stratospheric aerosols, but it is the difference between the two approaches which we concentrate on here.

**L307: Strange to use this argument here, when UKESM1 and WACCAM6 produce absolutely different climates with**

**strongly different AMOC intensity.**

As noted above, we disagree with the Reviewer that the climates of the two models at the end of the 21$^{st}$ century are "absolutely different" – certainly they differ, but that is what one would expect from two completely different climate models.

**L359: It is not only the amount of cooling that matters but a change in circulation**

We agree and have changed this sentence to reflect this (lines 362-364):

"We therefore focus our attention on the magnitude of the SAI-induced feedbacks on precipitation from the positive NAO anomaly compared with the temperature- and circulation-induced feedbacks on precipitation from global warming over the European area."

**L378: It was not observed but suggested. The question is still open.**

Agreed; we have changed the text as follows (lines 410-413):

"This is consistent with the form of SAI simulated in G6sulfur being essentially equivalent to a continuous large volcanic eruption in the tropics and indicates that the response to any putative continuous large-scale SO$_2$ injection is likely to be the same as that which has been suggested follows large sporadic eruptions."

**L381-384: This is a misinterpretation. Stenchikov et al. (2002) stated that along with the stratospheric mechanism (suggested by Graf and Kodera), the polar cooling due to polar ozone depletion and tropospheric planetary wave response could contribute in the positive NAO response.**

**There is a vast literature on NAO response to solar forcing (see Kodera's papers). It would be useful to compare the G6solar responses with that results. That would be a test for the models. There is no discrepancy (as for volcanic sulfate aerosols) between solar geoengineering forcing and natural forcing. Because in both cases, solar forcing is stationary.**

We agree that we were too simplistic in our summary of the results of Stenchikov *et al*. (2002). As these two sentences are something of an aside and rather tangential to the main objective of the paper, we have removed them from the revised manuscript (deleted lines 801-805 in the "tracked changes" file below).

A discussion of previous work on the response of the NAO to solar variability is now included - see our response to "General concerns" above. A comparison of model responses in the G6solar experiment with previous work would indeed be an interesting study, but that would be a completely different paper and beyond the scope of the present work.

Response to Reviewer #2

We thank the Reviewer for their comments. In our responses below, the Reviewer's comments are reproduced in **bold** text.

**General comments**

**This paper uses data from GEOMIP, where they compare two different solar radiation management techniques. One is a decreasing solar constant (G6Solar) and one is injection so2 in the stratosphere (G&Sulfur). In these experiments warming in the SSP5-8.5 scenario is reduce to SSP2-4.5. The author compares results from two climate models UKESM1 and WACCM6. The tittle of the paper is "North Atlantic Oscillation response in GeoMIP experiments**

**G6solar andG6sulfur: why detailed modelling is needed for understanding regional implications of solar radiation management", however the main results are not clearly connecting NAO to the observed changes. Main results are temperature, precipitation, and mean sea surface pressure response to different SRM techniques, and author does not show clearly how these responses are depending on the phase of NAO. This paper highlight the importance of atmospheric dynamical response to different SRM techniques and especially aerosols dynamical resposen.**

The Reviewer appears to have misunderstood the paper. We are very clear in the title that we are examining the effects of two different forms of simulating solar radiation management (SRM) on the NAO (and subsequent impacts). Despite this, the Reviewer appears to believe that we are examining the reverse of this, *i.e.* the effect of different phases of the NAO on SRM impacts. In addition to the title, we now make our objectives clearer in lines 111-113 where we say:

"The main objective is to determine whether, under SRM strategies which are continuous rather than sporadic or periodic in nature, the two models produce NAO responses that are consistent with the expectations discussed above: that SAI induces a significant shift to the positive phase of the NAO compared with reducing the solar constant."

In response to the Reviewer's statement that **"the main results are not clearly connecting NAO to the observed changes"** we would point out that throughout the paper we follow previous work on the NAO in climate models with particular reference to stratospheric aerosols (as used in the G6sulfur experiment but not in G6solar). Many of these studies have modelled the impacts of explosive volcanic eruptions and much of our work is guided by these earlier works which are fully referenced. After setting the scene in Figures 1-3 we use the following sequence of figures to connect stratospheric aerosol injection (SAI) SRM to NAO impacts, concentrating on the difference between G6sulfur (which includes SAI) and G6solar (which does not): Figure 4: SAI-induced perturbations to December-February (DJF) stratospheric temperature.

Figures 5 & 6: Resulting perturbations to the DJF zonal-mean stratospheric winds (as in Plate 5 of Shindell *et al*., 2001, cited

375 times), changes to the circumpolar jet at 10 hPa, and the subsequent perturbation to DJF 850 hPa zonal-mean wind.

Figure 7: The resulting perturbation to mean-sea level pressure distributions, as the NAO is defined in terms of the pressure difference between Iceland (low pressure) and the Azores (high pressure).

Figure 8: The impact on mean DJF near-surface air temperature, as in Figs. 2-5 of Shindell *et al.* (2004, cited 225 times) and Figs. 2 & 5 of Stenchikov *et al.* (2002, cited 233 times).

Figures 9, 10 & 11: The corresponding impact on DJF precipitation rate for the Northern Hemisphere and Europe.

As in previous work (which we reference), this logical trail of evidence connects the presence of stratospheric SRM aerosol to its effects on stratospheric temperature, then to stratospheric winds, through to tropospheric winds and a modification to the surface pressure distribution via an induced positive NAO and its resulting impacts on surface temperature and precipitation at hemispheric and European scales. We therefore respectfully reject the Reviewer's criticism that "**the main results are not**

**clearly connecting NAO to the observed changes.**"

**Selected model are quite different from each other, UKESM1 goes up to 85km where WACCM6 goes to 140km. How this affects to the results?**

The tops of the model do indeed differ, but such differences in model structure are to be expected in a model intercomparison project and are indeed part of the point of doing such a project. Although a detailed comparison between the two models is not the point of this study, the manuscript shows that the response of the northern hemisphere wintertime NAO to stratospheric aerosol injection compared with that to solar reduction is very similar in the two models, and that is the key metric of this study.

**Also author should include what aerosol-cloud proses are included in these models.**

The aerosol-cloud microphysical processes are described in the model description papers referred to in the manuscript. Changing the solar constant in the G6solar experiment does not directly affect aerosol or cloud microphysics, and the $SO_2$ injection in the G6sulfur experiment is in the stratosphere. We therefore believe that aerosol-cloud interactions, which are dominated by warm processes in the lower troposphere, are irrelevant here and to describe them here would detract from the message of the paper.

**This manuscript miss clear definition of NAO, Author should include the formula that they used to calculate NAO.**

We both define the NAO in words (lines 69-71) and describe it mechanistically (lines 75-82). We give a detailed description of the two different ways of quantifying the NAO, both as a simple difference in mean sea-level pressure between two points (lines 273-275) and by constructing an index which we fully explain (lines 275-279). We believe this is sufficient.

**Also author refer to different phase of NAO in the text, example in line 314. I recommend to included figures where the responses i.e for precipitation is shown separately for NAO positive and negative phase.**

The response of precipitation to different phases of the NAO is not the point of this study. As the title indicates, and as the revised manuscript now clearly states in the Introduction (lines 111-117), the point of the study is to examine whether geoengineering via stratospheric aerosol injection causes a shift to the positive phase of the NAO, something which would be missed if geoengineering was simply modelled as a reduction of the solar constant. The impact of such a positive shift of the NAO on Northern Hemisphere and European precipitation is then examined in sections 4.8 and 4.9. However, precipitation responses to positive and negative NAO phases is of no relevance to this study.

**In the result sections line 146 definition of present day run is not clear, it has been stated that PD is mean of 2011 - 2030, however what ssp scenario is used here is unclear.**

In the original manuscript the construction of the PD data was explained in the caption to Fig.1 which was being introduced at this point, but following the Reviewer's comment we have moved the following explanation into the main text (lines 176-
177):

"PD data are from years 2011-2014 of each model's CMIP6 historical experiment combined with years 2015-2030 from the corresponding ssp245 experiment."

**For reader it would be helpfull if all results are also showed respect to the present day.**
As explained in lines 111-117, the point of our paper is to examine the difference in impact on the NAO when geoengineering is modelled in detail (G6sulfur) compared to when it is modelled in an idealised manner (G6solar). The relevant comparison is therefore between G6sulfur and G6solar at the end of the 21$^{st}$ century, not between the end of the century and present day. We do present some differences compared with present-day (Figures 1, 9 and 10) but these are purely to provide some context for the differences between G6sulfur and G6solar.

**Specific comments**

**Line 23: In Abstract author should include model names**

We have now included the names of both models in the Abstract (line 23).

**Line 25: In abstract when author refers regional warming, spesifiy whitch regions.**

We have now amended the text as follows to be clearer about the regions concerned (line 26):

"…impacting the Eurasian continent leading to high-latitude warming over Europe and Asia."

**Line 26: "These findings are broadly consistent with previous findings on the impact of stratospheric volcanic aerosol**
**on the NAO" specify this. What are the previous findings**

As we state in the manuscript, the "previous findings" are similar to those found in our study which we described in the immediately preceding sentence (lines 24-26 in the revised manuscript). To try to make this clearer we have revised the text as follows (lines 26-27):

"These results are broadly consistent with previous findings which show similar impacts from stratospheric volcanic aerosol
on the NAO…"

**Line 36: author talks about aerosol-cloud interactions, author should specify the different interactions mechanisms.**

As noted above, in our opinion the details of aerosol-cloud interactions are irrelevant for this study. They are mentioned here simply as part of a general explanation of why aerosols are considered important to the radiative forcing of the Earth's climate.

**Line 70: This sections deals of definitions of NAO. This should be in Method section.**

We have to disagree with the Reviewer. An explanation of the NAO is not an experimental method but an important part of the introduction so that the reader can understand the basic physical processes in play.

**Line 85-90: Deals with model selections, this should also be in method sections and include some arguments are the model independent**

We are sorry to again disagree with the Reviewer, but these lines (now lines 104-111 in the revised manuscript) form a simple introduction to the experiments under discussion and in the revised manuscript form a lead-in to a more specific description of the objectives of the study (lines 111-120). A more detailed explanation of the experimental design in presented in section 3.

We note that Reviewer #1 found the paper "well written and logically organized" and we are therefore loath to change the construction of the paper.

**Line 144: Define key variables**

The key variables are all presented in a logical sequence in the sub-sections which immediately follow this line in section 4 (line 171 in the revised manuscript), each sub-section having a title which clearly describes the quantity discussed, so we see no need to include a list of these variables at this point.

**Line 151: Include the difference picture**

As we state in the manuscript (lines 181-183) the point here is that the inter-model differences for a given forcing are much larger than the inter-forcing differences for a given model, *i.e.* all the panels on the top line look very similar to each other, as do all the ones on the bottom line, and that the two lines look different. The actual details of the inter-model differences (*i.e.* between the upper and lower rows of Fig, 1) have no relevance to this discussion and including an extra figure to show them would be an unnecessary distraction from the point being made.

**Line 185: Include more**

We are not sure whether this comment was truncated in the reviewer response and we cannot identify any problem on this line (now line 214).

[revised manuscript text omitted]

---

## Referee Report (RR1)

**North Atlantic Oscillation response in GeoMIP experiments G6solar and G6sulfur: why detailed modelling is needed for understanding regional implications of solar radiation management**

Authors have made the text much cleaner. Authors have responded clearly to all review comments. I will recommend to accept this version of the paper. The paper broadens our understanding of geoengineering

---

## Author Response (AR2)

**Response to Editor's comments**

We thank the Editor for his comments and suggestions (reproduced below in bold). Our responses are as follows:

**1. L174-176 is about the fact that the geoengineering perturbation is much stronger than a Pinatubo-like perturbation. You have simply responded that it has to be -- perhaps a more useful comment to add to the paper would be to add this explicitly -- making the point that whilst drawing some analogy with e.g. Pinatubo-like impacts is useful, but has limitations (with the amplitude of the perturbation being one of them).**

We have inserted the following text at lines 305-321 in the tracked-changes version attached below. We now explicitly note the large size of the perturbation compared with Pinatubo (and the possible limitations of using observed eruptions as analogues) and attempt to place the magnitude of these geoengineering emissions in context. We have also modified the title of this subsection (4.2) to reflect the increased discussion of $SO_2$ injection rates.

"Such injection rates are broadly similar to the amount injected by the 1991 eruption of Mt. Pinatubo (Guo *et al*., 2004) but unlike the latter they continue year on year. Such large, persistent perturbations are obviously different to the pulse-like injection and subsequent exponential decay of explosive volcanic eruptions (*e.g*., Jones *et al*., 2016a) which suggests that one cannot simply assume that the responses to such SAI would be analogous to those from volcanic eruptions. The injection rates by the end of the century have to be so large to counteract the warming due to the increased concentration of
atmospheric carbon dioxide which has accumulated over the period 1850-2100. While such injection rates appear high, they are typical in model geoengineering studies. A previous GeoMIP experiment known as G3 (Kravitz *et al*., 2011) involved injecting increasing amounts of $SO_2$ to offset anthropogenic radiative forcing in the RCP4.5 scenario (Thomson *et al*., 2011) over the period 2020-2070 and Niemeier *et al*. (2013) found that an injection rate of around 12 Tg of $SO_2$ per year was needed in their model by 2070. Niemeier and Timmreck (2015) suggested a massive 90 Tg of $SO_2$ per year would be needed
by 2100 to offset the temperature change in the RCP8.5 scenario (Riahi *et al.*, 2011) in a model that explicitly simulated the evolution of aerosol microphysics to larger sizes via condensation and coagulation as the injection rate increased. The increase in aerosol size leads to a decreased cooling efficiency per unit mass with increasing $SO_2$ owing to decreased stratospheric lifetime (caused by higher aerosol terminal velocities) and also less efficient cooling in the shortwave part of the spectrum along with a stronger counterbalancing impact on terrestrial radiation (Niemeier and Timmreck, 2015). Both
UKESM1 and CESM2-WACCM6 include these microphysical mechanisms, so the injection rates used here are by no means exceptional in SAI geoengineering studies."

Additional references:

Guo, S., Bluth, G. S., Rose, W. I., Watson, I. M., and Prata, A. J.: Re-evaluation of the $SO_2$ release of the 15 June 1991 Pinatubo eruption using ultraviolet and infrared satellite sensors, *Geochem. Geophys. Geosyst.*, **5**, Q04001, doi:10.1029/2003GC000654, 2004.

Jones, A. C., Haywood, J. M., Jones, A., and Aquila, V.: Sensitivity of volcanic aerosol dispersion to meteorological
conditions: A Pinatubo case study, *J. Geophys. Res.*, **121**, 6892–6908, doi:10.1002/2016JD025001, 2016a.

Niemeier, U., Schmidt, H., Alterskjær, K., and Kristjánsson, J. E.: Solar irradiance reduction via climate engineering: Impact
of different techniques on the energy balance and the hydrological cycle, *J. Geophys. Res.*, **118**, 11905–11917, doi:10.1002/2013JD020445, 2013.

Niemeier, U., and Timmreck, C: What is the limit of climate engineering by stratospheric injection of $SO_2$? *Atmos. Chem. Phys.*, **15**, 9129-9141, doi:10.5194/acp-15-9129-2015, 2015.

Riahi, K., Rao, S., Krey, V., Cho, C., Chirkov, V., Fischer, G., Kindermann, G., Nakicenovic, N., and Rafaj, P.: RCP 8.5 - A scenario of comparatively high greenhouse gas emissions, *Climatic Change*, **109**, 33–57, doi:10.1007/s10584-011-0149-y, 2011.

Thomson, A. M., Calvin, K. V., Smith, S. J., Kyle, G. P., Volke, A., Patel, P., Delgado-Arias, S., Bond-Lamberty, B., Wise, M. A., Clarke, L. E.., and Edmonds, J. A.: RCP4.5: A pathway for stabilization of radiative forcing by 2100, *Climatic Change*, **109**, 77–94, doi:10.1007/s10584-011-0151-4, 2011.

2.  **L202-205 Perhaps the referee is looking for some comment that this large temperature change is not unexpected given the large aerosol perturbation -- e.g. perhaps it is consistent with a simple scaling up of a Pinatubo like perturbation. You have made the point that the models being used have been validated in Pinatubo-like scenarios -- so perhaps no further comment is needed.**
We believe that we have said what is required on this point and agree with the Editor that no further comment is needed.

3.  **L307: The referee's comment was 'Strange to use this argument here, when UKESM1 and WACCAM6**
**produce absolutely different climates with 205 strongly different AMOC intensity.' about your sentence**

**'Reasons for these differences are beyond the scope of this work but demonstrate that important inter-model differences still exist in state-of-the-art climate models.'. Perhaps that sentence at this particular point in the paper doesn't achieve very much. From my point of view the key overall point is that where your two model results disagree there is fundamental uncertainty -- you have said that at the end of your abstract and perhaps**
**it should be re-iterated in the final section.**

We have deleted the "Reasons for these differences…" sentence from the end of Section 4.7 (lines 456-457 below) and have added the following text to the end of the penultimate paragraph in the Conclusions section (lines 547-549):

[revised manuscript text omitted]